# DOES SGD REALLY HAPPEN IN TINY SUBSPACES?

**Minhak Song**
KAIST Math

**Kwangjun Ahn**
Microsoft Research

**Chulhee Yun**
KAIST AI

## ABSTRACT

Understanding the training dynamics of deep neural networks is challenging due to their high-dimensional nature and intricate loss landscapes. Recent studies have revealed that, along the training trajectory, the gradient approximately aligns with a low-rank top eigenspace of the training loss Hessian, referred to as the dominant subspace. Given this alignment, this paper explores whether neural networks can be trained within the dominant subspace, which, if feasible, could lead to more efficient training methods. Our primary observation is that when the SGD update is projected onto the dominant subspace, the training loss does not decrease further. This suggests that the observed alignment between the gradient and the dominant subspace is spurious. Surprisingly, projecting out the dominant subspace proves to be just as effective as the original update, despite removing the majority of the original update component. We observe similar behavior across practical setups, including the large learning rate regime (also known as Edge of Stability), Sharpness-Aware Minimization, momentum, and adaptive optimizers. We discuss the main causes and implications of this spurious alignment, shedding light on the dynamics of neural network training.

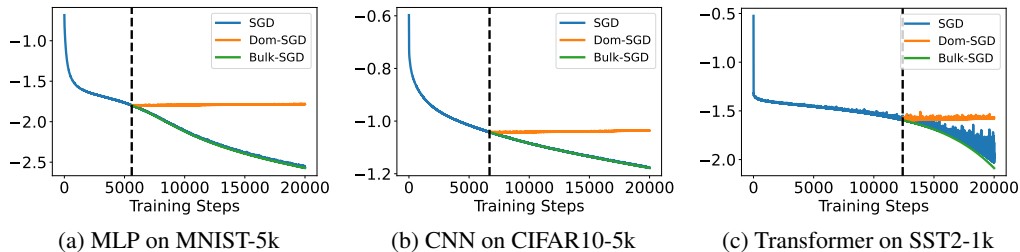

| (a) MLP on MNIST-5k | (b) CNN on CIFAR10-5k | (c) Transformer on SST2-1k |

Figure 1: The summary of our main results in Section 3 (training loss in log-scale). For neural network training, Gur-Ari et al. (2018) observe that gradients approximately align with the dominant subspace, spanned by the dominant eigenvectors of the training loss Hessian. To see whether such phenomenon lets us train neural networks within the dominant subspace, we implement Dom-SGD, where each SGD update is projected onto the dominant subspace. Surprisingly, training stops after this modification, suggesting that **the dominant subspace is not where the learning happens**. In contrast, Bulk-SGD, where we project each SGD updates onto the bulk subspace orthogonal to the dominant subspace, is just as effective as the original update, despite removing the majority of original updates. Experimental details are provided in Appendix B.

## 1 INTRODUCTION

Understanding the optimization of deep neural networks presents a complex challenge, given their high-dimensional nature and the intricate characteristics of their training loss landscapes. Over the last decade, an abundance of studies has investigated the landscape of training loss $L : \mathbb{R}^p \to \mathbb{R}$ (Li et al., 2018b; He et al., 2019). In this work, we are interested in the following noteworthy phenomena:

- **Hessian is approximately low-rank.** Extensive research (Sagun et al., 2016; 2017; Ghorbani et al., 2019; Papyan, 2019; 2020) has revealed that for $k$-class classification problems, the loss Hessian $\nabla^2 L$ exhibits a "low-rank" structure, characterized by $k$ dominant eigenvalues significantly larger than the others. See Figure 2 for details.

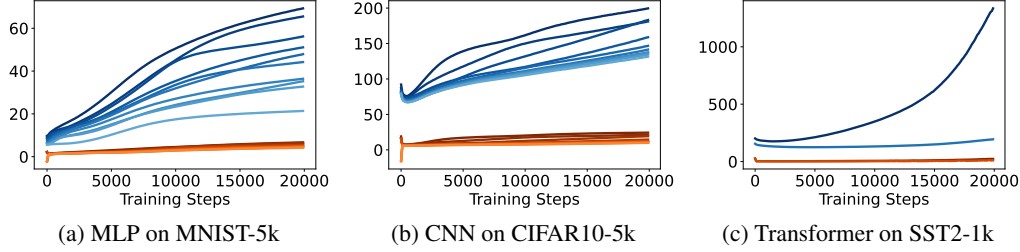

(a) MLP on MNIST-5k     (b) CNN on CIFAR10-5k     (c) Transformer on SST2-1k

Figure 2: **Low-rank structure of the Hessian.** The plot shows the top eigenvalues of the loss Hessian during SGD training. The blue curves represent the top-$k$ eigenvalues, which are significantly larger than the next top-$k$ eigenvalues, shown in orange. Here, $k$ corresponds the number of classes in the classification task ($k = 10$ for MNIST-5k, CIFAR10-5k, and $k = 2$ for SST2-1k).

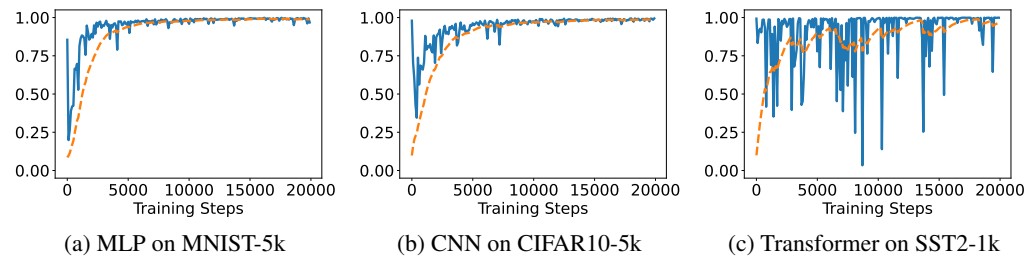

(a) MLP on MNIST-5k     (b) CNN on CIFAR10-5k     (c) Transformer on SST2-1k

Figure 3: **Alignment of gradients with dominant subspaces.** The plot illustrates $\chi_k(\nabla L(\theta_t))$ during SGD training, where $k$ is the number of classes for the classification task (see Definition 2). The orange dashed lines represent the exponential moving average (EMA) of $\chi_k(\nabla L(\theta_t))$. After a few early steps, $\chi_k(\nabla L(\theta_t))$ reaches and stays near 1, indicating the alignment between gradients and dominant subspaces.

- **Gradients approximately align with the low-rank eigenspace.** Gur-Ari et al. (2018) demonstrated that during SGD training, gradients tend to align closely with the low-dimensional subspace spanned by the dominant eigenvectors of the loss Hessian. See Figure 3 for details.

We provide a more extensive background on these phenomena in Appendix A. The eigenspace of the top-$k$ eigenvalues of $\nabla^2 L(\theta)$, referred to as the **dominant subspace** at $\theta$, is the main focus of this work. Motivated by Gur-Ari et al. (2018), we ask:

> *Q. Can deep neural networks be trained within the dominant subspace?*       (Q1)

This question carries significant practical implications, potentially leading to more efficient training methods for neural networks. Furthermore, it offers insights into why deep learning optimization may not suffer from the curse of dimensionality despite operating in high-dimensional spaces.

## 1.1 SUMMARY OF MAIN RESULTS

In this paper, we rigorously examine the question (Q1) through systematic experiments. Quite surprisingly, our results reveal that the answer to the question is negative, as summarized below.

- In Section 3, we demonstrate that the observed alignment is "spurious" in the sense that the aligned component of the gradient is not beneficial for training, even though it constitutes the majority of the gradient. Specifically, we run a critical experiment where we modify SGD by projecting each update onto the *dominant subspace*; we call this Dom-SGD. Unexpectedly, Dom-SGD does not further decrease the training loss. Next, we consider Bulk-SGD which projects each update onto the *bulk subspace*, *i.e.*, the orthogonal complement of the dominant subspace. Despite the fact that the major component of each update is removed, this time the training remains as effective as the original update (see Figure 1).
- In Section 4, we identify that the spurious alignment between the gradient and the dominant subspace is caused by the stochastic noise inherent to SGD in the gradient flow (GF)

regime, by showing that alignment disappears when using full-batch GD (see Figure 4 and Figure 5). Additionally, we present a simple quadratic model which also captures the observed phenomena, providing insights into the role of stochastic noise in the spurious alignment (see Figure 6 and Figure 7).

- In Section 5, we extend our observations to two other practical settings: (1) GD in the Edge of Stability (EoS) regime (Cohen et al., 2021), and (2) Sharpness-Aware Minimization (SAM) (Foret et al., 2021). For both of the settings, we again observe that each update approximately aligns with the dominant subspace, yet the aligned component of each update does not contribute to the loss decrement (see Figure 9 and Figure 10). For GD in the EoS regime, the alignment is due to the self-stabilization mechanism (Damian et al., 2023), in contrast to SGD in the GF regime where stochastic noise is the cause of the alignment.

- In Section 6, we again observe that for momentum and adaptive optimizers, *e.g.* Adam, if we project the update vector onto the dominant subspace, the training loss fails to decrease. Moreover, we demonstrate that momentum and adaptive methods amplify the bulk subspace component of each update, partially explaining their success in neural network training (see Table 1 and Figure 11).

## 2 STARTING POINT: GRADIENT ALIGNS WITH THE DOMINANT SUBSPACE

In this section, we set the stage for our main results by reviewing the main observation of Gur-Ari et al. (2018). To that end, we first introduce some notations to ease our discussion.

**Notations.** Let $[n]$ denote the set $\{1, 2, \ldots, n\}$. For the Hessian to be well-defined, let $L : \mathbb{R}^p \to \mathbb{R}$ be a twice-differentiable training loss. For $\theta \in \mathbb{R}^p$, let $\lambda_1(\theta), \lambda_2(\theta), \ldots, \lambda_p(\theta)$ denote the eigenvalues of the loss Hessian $\nabla^2 L(\theta) \in \mathbb{R}^{p \times p}$ in descending order, and let $u_1(\theta), u_2(\theta), \ldots, u_p(\theta)$ denote the corresponding eigenvectors. Given these notations, we begin with the most important concept for our discussion, namely, the *dominant subspace*.

**Definition 1 (Dominant subspace).** *The top-$k$ **dominant subspace** at $\theta$ is denoted by*

$$\mathcal{S}_k(\theta) := \mathrm{span}\{u_i(\theta) : i \in [k]\},$$

*and its orthogonal complement by $\mathcal{S}_k^{\perp}(\theta)$, referred to as the **bulk subspace**. Unless specified otherwise, the default choice for $k$ is the number of classes for the classification task.*

**Definition 2 (Dominant subspace projection).** *The projection matrix onto the dominant subspace $\mathcal{S}_k(\theta)$ is denoted by*

$$P_k(\theta) := \sum_{i=1}^{k} u_i(\theta) u_i(\theta)^{\top} \in \mathbb{R}^{p \times p},$$

*and the projection matrix onto $\mathcal{S}_k^{\perp}(\theta)$ by $P_k^{\perp}(\theta) := I - P_k(\theta)$. For a given vector $v \in \mathbb{R}^p$, we can decompose the vector into $v = P_k(\theta)v + P_k^{\perp}(\theta)v$. We say $P_k(\theta)v$ is the **dominant component** of $v$, and $P_k^{\perp}(\theta)v$ is the **bulk component** of $v$. We denote the fraction of $v$ in the dominant subspace by*

$$\chi_k(v; \theta) := \|P_k(\theta)v\|_2 / \|v\|_2,$$

*with $\chi_k(v; \theta) = 0$ if $\|v\|_2 = 0$. A vector $v \in \mathbb{R}^p$ is said to (approximately) align with the dominant subspace $\mathcal{S}_k(\theta)$ if $\chi_k(v; \theta)$ is close to 1, and align with $\mathcal{S}_k^{\perp}(\theta)$ if $\chi_k(v; \theta)$ is close to 0.*

When clear from context, we use shorthand notation such as $\lambda_i := \lambda_i(\theta)$, $u_i := u_i(\theta)$, $\nabla L := \nabla L(\theta)$, $\mathcal{S}_k := \mathcal{S}_k(\theta)$, $\mathcal{S}_k^{\perp} := \mathcal{S}_k^{\perp}(\theta)$, $P_k := P_k(\theta)$, $P_k^{\perp} := P_k^{\perp}(\theta)$, and $\chi_k(v) := \chi_k(v; \theta)$.

Using our notations, the striking observation of Gur-Ari et al. (2018) can be formalized as follows.

**Phenomenon 1. Gradient approximately aligns with the dominant subspace along SGD trajectories.** *Consider the SGD trajectory $\{\theta_t\}$ with a constant learning rate. After a few initial steps, $\chi_k(\nabla L(\theta_t))$ quickly reaches and remains near 1.*

In Figure 3, we confirm the main results of Gur-Ari et al. (2018) for various settings. Notice that $\chi_k(\nabla L(\theta_t))$ reaches 1 after a few early steps, indicating that the gradient $\nabla L(\theta_t)$ approximately aligns with the dominant subspace $\mathcal{S}_k(\theta_t)$. Given this alignment, it seems that the training can be done within the dominant subspace, which leads to the previously introduced question (Q1). In the next section, we conduct a set of experiments to investigate (Q1).

## 3 NEURAL NETWORKS CANNOT BE TRAINED WITHIN DOMINANT SUBSPACES

In this section, we present the first main observation of this paper regarding question (Q1). We start with a preliminary analysis using the local quadratic approximation of the neural network landscape.

### 3.1 WHAT DO WE EXPECT BASED ON QUADRATIC TAYLOR APPROXIMATION?

To analyze the convergence of gradient-based optimization algorithms, a common approach is to use the local quadratic Taylor approximation (see, e.g., (Ghadimi & Lan, 2013)). The "descent lemma" characterizes the one-step progress of the optimizer $L(\theta_{t+1}) - L(\theta_t)$ using this approximation, assuming the training loss $L$ is smooth. Based on the local quadratic Taylor approximation, we have:

$$L(\theta_{t+1}) - L(\theta_t) \approx \underbrace{\langle \nabla L(\theta_t), \theta_{t+1} - \theta_t \rangle}_{=:\text{gradient correlation}} + \frac{1}{2}(\theta_{t+1} - \theta_t)^\top \nabla^2 L(\theta_t)(\theta_{t+1} - \theta_t). \tag{1}$$

Let us denote the first term on the RHS by the *gradient correlation* term and the second term by the *second-order error* term. Since the SGD updates are defined as

$$\theta_{t+1} \leftarrow \theta_t - \eta g_t$$

for the stochastic gradient $g_t$ at $\theta_t$, the gradient correlation term is negative in expectation:

$$\mathbb{E}[\text{gradient correlation}] = \mathbb{E}[\langle \nabla L(\theta_t), -\eta g_t \rangle] = -\eta \|\nabla L(\theta_t)\|^2 < 0. \tag{2}$$

For the experiments in Figures 1a and 1b, we use small learning rates to ensure SGD closely follow the continuous-time gradient flow so that the training loss decreases nearly monotonically, suggesting that the negative gradient correlation dominates the second-order error term in these cases.

Hence, if the quadratic Taylor approximation was accurate enough, based on the above analysis and Phenomenon 1, it is expected that one can decrease the training loss based on updates lying in the dominant subspace, as hypothesized in the question (Q1).

To directly test this hypothesis, we design the following critical experiment.

> **Our critical experiment:** In the same settings as before, whenever Phenomenon 1 occurs, consider the following updates where each update of SGD is projected onto the dominant subspace:
>
> $$\theta_{t+1} \leftarrow \theta_t - \eta P_k(\theta_t) g_t. \tag{Dom-SGD}$$

By Phenomenon 1, Dom-SGD has an approximately same gradient correlation as SGD given in (2):

$$\mathbb{E}[\text{gradient correlation}] = \mathbb{E}[\langle \nabla L(\theta_t), -\eta P_k(\theta_t) g_t \rangle] \approx -\eta \|\nabla L(\theta_t)\|^2.$$

Therefore, based on the local quadratic Taylor approximation, Dom-SGD should be able to successfully train neural networks whenever Phenomenon 1 occurs. Is it really the case?

### 3.2 THE "SPURIOUS" ALIGNMENT WITH THE DOMINANT SUBSPACE

In the same settings as before, we first train neural networks with SGD up until we observe Phenomenon 1. Specifically, we track the exponential moving average (EMA) of $\chi_k(\nabla L(\theta_t))$ values (EMA factor set to 0.9), and switch from SGD to Dom-SGD whenever the EMA value exceeds 0.95. Note that we recompute the dominant subspace at every step when running Dom-SGD.

For various settings, we plot the training loss of Dom-SGD in Figure 1, comparing it with SGD under the same initialization. We employ a constant learning rate and mean squared error (MSE) loss for classification (Hui & Belkin, 2021; Cohen et al., 2021). Additional experiments, including those using cross-entropy loss and training standard architectures, are provided in Appendix C.

Consistently, we observe that **Dom-SGD fails to further decrease the training loss**, unlike standard SGD. Actually, we observe that Dom-SGD even slowly increases the loss in long run. This suggests that the *dominant component* of the stochastic gradient $g_t$ is in fact not beneficial for training, despite constituting the majority of $g_t$. We refer to the alignment between the gradient and the dominant subspace in this scenario as "spurious," because, based on the local quadratic Taylor approximation in Section 3.1, we expect that projecting the update vector onto the dominant subspace should decrease the loss similarly to the original update if the update vector is aligned with the dominant subspace.

**Remark 1** (This is **not the end-of-training phenomenon**). *Here, we note that the switching happens when training accuracy reaches around $53\%$ for CNN on CIFAR10-5k, and $69\%$ for Transformer on SST2-1k. This indicates that the observed phenomenon is not confined to the SGD dynamics near the manifold of local minimizers, which is the basis of many recent theoretical analyses (see, e.g., (Arora et al., 2022; Li et al., 2022; Lyu et al., 2022; Wen et al., 2023; Ahn et al., 2024b)).*

### 3.3 BULK SUBSPACE IS WHERE THE LEARNING HAPPENS

To further strengthen our main observation, we conduct another set of experiments, wherein we switch from SGD to the following update scheme:

$$\theta_{t+1} \leftarrow \theta_t - \eta P_k^\perp(\theta_t)g_t \,. \tag{Bulk-SGD}$$

Essentially, Bulk-SGD discards the majority of the stochastic gradient $g_t$ by removing its dominant component. Consequently, it seems less likely that the remaining bulk component of stochastic gradient would lead to successful training.

Surprisingly, as shown in Figure 1, **Bulk-SGD is as effective as SGD in decreasing the training loss**. This further highlights that it is indeed a small fraction of gradient that aligns with the bulk subspace that contributes to training loss decrease.

One can summarize our observation thus far as follows.

**Phenomenon 2.** *Although the gradient approximately aligns with the dominant subspace at each step, **the training loss does not decrease within the dominant subspace**, suggesting a "spurious" alignment. Surprisingly, with Bulk-SGD, where each update is projected onto bulk subspaces, the training remains as effective as the original update. This emphasizes the importance of a small component of the update that aligns with the bulk subspace.*

Based on our preliminary analysis in Section 3.1, Phenomenon 2 appears quite counterintuitive and unexpected. The next section focuses on explaining how this counterintuitive phenomenon occurs. In particular, the seemingly contradictory conclusion from Section 3.1 will be revisited in Section 4.3.

## 4 WHAT CAUSES THE SPURIOUS ALIGNMENT WITH DOMINANT SUBSPACES?

In this section, we aim to explain the spurious alignment discussed in Phenomenon 2. To that end, we first distinguish between two different regimes of SGD dynamics, because the underlying mechanism of the alignment are different.

**Definition 3.** *We say (S)GD is in the **GF regime** when the sharpness is below the maximum stable sharpness (MSS),[1] where it closely follows the gradient flow. In the GF regime, the sharpness typically increases (progressive sharpening), and the loss decrease is stable. Conversely, (S)GD is in the **EoS regime** when the sharpness is close to MSS. In the EoS regime, the sharpness oscillates around MSS, and the loss decrease is spiky and unstable (Cohen et al., 2021).*

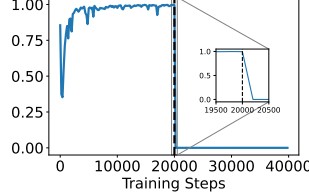

Figure 4: $\chi_k(\nabla L(\theta_t))$ when switching from SGD to GD at step 20000 while training MLP on MNIST-5k.

In this section, we focus on SGD in the GF regime and present the mechanism of how gradients align with the dominant subspace. The scenario of the EoS regime will be discussed in Section 5.1. For SGD in the GF regime, the spurious alignment is closely tied to the landscape of the training loss near SGD trajectories. Importantly, SGD introduces inherent *randomness* into its trajectories. We investigate how this *stochastic noise* affects the phenomenon.

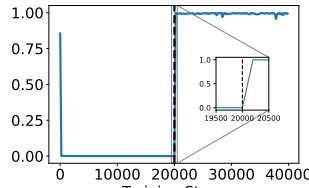

Figure 5: $\chi_k(\nabla L(\theta_t))$ when switching from GD to SGD.

### 4.1 STOCHASTIC NOISE OF SGD IS THE MAIN CAUSE

We examine the role of stochastic noise by contrasting the behavior of SGD with (full-batch) GD, isolating the effect of stochastic noise.

---

[1]MSS is $2/\eta$ for GD (Cohen et al., 2021), and smaller for SGD (Wu et al., 2018).

First, we demonstrate the crucial role of stochastic noise in the alignment by switching from SGD to GD when Phenomenon 1 is observed. Strikingly, as shown in Figure 4, the **alignment disappears as soon as we switch to GD**. More specifically, $\chi_k(\nabla L(\theta_t))$ quickly becomes 0 as soon as the switch occurs. This sharp transition indicates that the stochastic noise must play a crucial role.

In Section D.1, when training neural networks with GD (from scratch) in the GF regime, we observe **no alignment between gradients and dominant subspaces**, in contrast to SGD. In this case, $\chi_k(\nabla L(\theta_t))$ quickly reaches and remains near 0, indicating alignment of gradients with bulk subspaces. Remarkably, despite differences in gradient alignment (GD: $\chi_k(\nabla L) \approx 0$, SGD: $\chi_k(\nabla L) \approx 1$), GD and SGD trajectories closely track each other (see Section D.2). This suggests that the presence of small stochastic noise results in a drastically different behavior in the alignment.

In Figure 5, we switch our optimizer from GD to SGD, complementary to the experiment in Figure 4. This time we observe that the alignment sharply appears, *i.e.*, $\chi_k(\nabla L(\theta_t))$ quickly becomes 1, as soon as the switch occurs. Note that SGD is still in the GF regime as sharpness stably increases, rather than oscillating. This highlights that the gradient alignment during SGD is **not** due to self-stabilization effect (Damian et al., 2023) in the EoS regime. To sum up, one can summarize our findings as follows.

**Phenomenon 3** (**The spurious alignment is due to stochastic noise).** *In the GF reigme, the alignment between the gradient and the dominant subspace quickly disappears when switching the optimizer from SGD to GD. Moreover, the alignment quickly reappears when switching GD back to SGD. Hence, the spurious alignment is mainly due to the stochastic noise of SGD.*

To further demonstrate that noise is the primary driver of gradient alignment for SGD, we conduct experiments using Noisy Gradient Descent (NGD) and SGD with varying batch sizes in Section D.4. We implement NGD by injecting Gaussian noise after each GD update iteration. We observe that either increasing the noise scale or decreasing the batch size lead to the increased gradient alignment. This observation further demonstrates that noise is the main cause of the spurious alignment for SGD.

Given this observation, one might question how the presence of small stochastic noise leads to a drastically different alignment. We investigate this in the next subsection using a simple model.

## 4.2 Understanding the role of stochastic noise via a toy quadratic model

Towards understanding Phenomena 1–3, this section introduces a simple example that recovers all the phenomena.

Given the typical ill-conditioned nature of neural network training, we consider a 2-dimensional ill-conditioned quadratic loss, $L(x, y) = \frac{1}{2}(1000x^2 + y^2)$, where $\theta = (x, y) \in \mathbb{R}^2$. We define $\ell_1(x, y) = L(x, y) + 100xy$ and $\ell_2(x, y) = L(x, y) - 100xy$, resulting in $L(x, y) = \frac{1}{2}(\ell_1(x, y) + \ell_2(x, y))$.

We conduct GD with learning rate $\eta$ as

$$\theta_{t+1}^{\mathrm{GD}} \leftarrow \theta_t^{\mathrm{GD}} - \eta \nabla L(\theta_t^{\mathrm{GD}}),$$

and SGD using random sampling with the same learning rate $\eta$ as

$$\theta_{t+1}^{\mathrm{SGD}} \leftarrow \theta_t^{\mathrm{SGD}} - \eta \nabla \ell_k(\theta_t^{\mathrm{SGD}}), \quad \text{where } k \sim \mathrm{Unif}(\{1, 2\}).$$

In Figure 6, we visualize the optimization trajectories of GD and SGD with an initialization $\theta_0^{\mathrm{GD}} = \theta_0^{\mathrm{SGD}} = (1, 1)$ and a learning rate $\eta = 10^{-4}$. The Hessian of the quadratic loss remains constant during training, with eigenvalues $\lambda_1 = 1000$ and $\lambda_2 = 1$, and corresponding eigenvectors $e_1 = (1, 0)$ and $e_2 = (0, 1)$. We compute the fraction of gradient in the dominant subspace as $\chi_1(\nabla L(\theta)) := \frac{|\langle \nabla L(\theta), e_1 \rangle|}{\|\nabla L(\theta)\|_2}$, as shown in Figure 7.

Notably, **this simple quadratic model recovers all the observed phenomena** (Phenomena 1–3). In both GD and SGD trajectories, $x_t$ quickly converges to 0 due to the sharper direction along $e_1$ ($\lambda_1 \gg \lambda_2$). Subsequently, both trajectories remain close to the $y$-axis throughout the remaining of the training. However, in GD, $\chi_1(\nabla L(\theta_t^{\mathrm{GD}}))$ quickly approaches

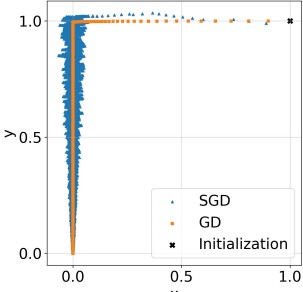

Figure 6: GD and SGD trajectories when training a 2-dimensional ill-conditioned toy quadratic model.

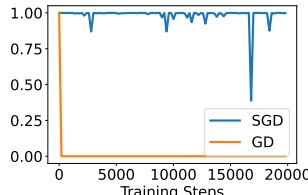

Figure 7: $\chi_1(\nabla L(\theta_t))$ during GD and SGD for Figure 6.

and remains near 0 (Phenomenon 3), while in SGD, $\chi_1(\nabla L(\theta_t^{\text{SGD}}))$ stays close to 1 (Phenomenon 1). Notice that if we run Dom-SGD, the updates will be done only in the $x$ direction, hence training stops after switching to Dom-SGD (Phenomenon 2). We provide results on NGD in Appendix D.6.

The discrepancy in alignment between GD and SGD arises from the ill-conditioned nature of the loss landscape, where the small stochastic noise of SGD in the $x$ direction induces a large gradient component along the $x$ direction. For example, if the SGD iterate is at $\theta = (0.01, 0.5)$, a slight departure from the $y$-axis, the gradient alignment $\chi_1(\nabla L)$ is approximately 0.999.

### 4.3 Revisiting our preliminary analysis (Section 3.1)

At this point, some readers might wonder how we reconcile the results with our preliminary analysis (Section 3.1). Based on our investigations so far, we propose one plausible explanation that the training loss landscape is locally "ill-conditioned-valley"-like. This landscape has two key features causing the spurious alignment:

- The landscape is locally valley-shaped, where it is steep along the dominant subspace and flat along the bulk subspace. In particular, the curvature along the dominant subspace is much larger than that along the bulk subspace.
- The bottom of the valley is connected along the bulk subspace. Moreover, there is a direction within the bulk subspace along which the bottom of the valley descends.

To aid readers' understanding, we provide a simple illustration of an ill-conditioned-valley-like landscape in Figure 8. With these features, Phenomena 1–3 can indeed occur:

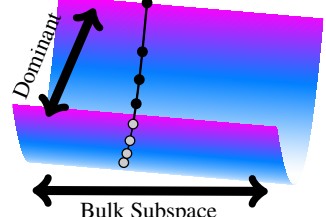

- Due to stochastic noise, the SGD iterates slightly deviate from the bottom of the valley. Subsequently, the high curvature along the dominant subspace causes gradients to align with this subspace, *i.e.*, the iterates exhibit Phenomenon 1.
- However, Dom-SGD fails to further decrease the training loss, as observed in Phenomenon 2, since it fails to follow the true progress direction along the bulk subspace.
- Moreover, without stochastic noise, the iterates quickly approach the bottom of the valley, where the alignment disappears, as described in Phenomenon 3.

Figure 8: Illustration demonstrating how spurious alignment can occur with an ill-conditioned-valley-like training loss. Dom-SGD iterates (depicted with dots) fail to progress along the bulk subspace where the training loss decreases.

In Section D.5, we measure the distance of the weights from the step where we switch from SGD to Dom-SGD and Bulk-SGD for the experiments in Figure 1. We observe that weights do not move far from the switching step for Dom-SGD, in contrast to SGD and Bulk-SGD. Furthermore, Dom-GD shows near-zero movement, suggesting it gets stuck at the bottom of the valley, consistent with our proposed explanation based on an ill-conditioned-valley-like landscape.

Given our results for SGD thus far, one natural question is whether these phenomena are also observed for other practical optimization algorithms.

## 5 Edge of Stability and Sharpness-Aware Minimization

In this section, we extend our investigations to two other practical settings: (1) (S)GD in the Edge of Stability (EoS) regime (Cohen et al., 2021), and (2) Sharpness-Aware Minimization (SAM) (Foret et al., 2021). We show that the same phenomena are observed for the two settings: the update direction aligns with the dominant subspace, but the alignment is again spurious.

### 5.1 Edge of Stability

Recent empirical studies (Jastrzębski et al., 2020; Cohen et al., 2021) have observed that when training neural networks using full-batch GD with large learning rates $\eta$, the sharpness $\lambda_1$ increases until it

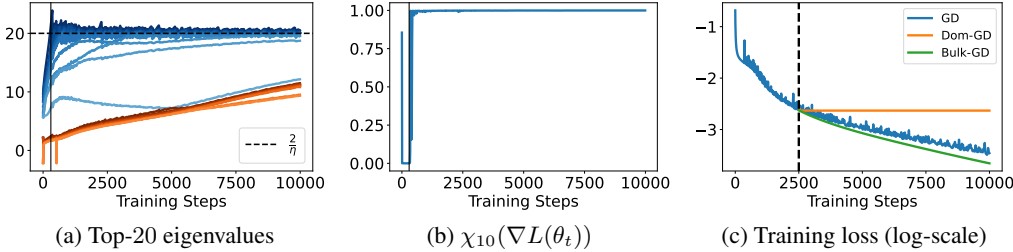

(a) Top-20 eigenvalues          (b) $\chi_{10}(\nabla L(\theta_t))$          (c) Training loss (log-scale)

Figure 9: **Gradients approximately align with dominant subspaces in the EoS regime.** Training MLP on MNIST-5k with GD using a large learning rate $\eta = 0.1$. (a) The plot shows the top-10 eigenvalues in blue and the next top-10 eigenvalues in orange. After a few steps, GD enters the EoS regime, where the sharpness stabilizes near $2/\eta$. (b) As the sharpness reaches $2/\eta$, $\chi_{10}(\nabla L(\theta_t))$ shoots up and remains near 1. (c) We switch the optimizer from GD to Dom-GD and Bulk-GD at step 2500. **Dom-GD fails to further decrease the training loss, in contrast to GD and Bulk-GD.**

reaches the stability threshold, or the maximum stable sharpness (MSS), $2/\eta$, and saturates around the threshold (see Figure 9a). Cohen et al. (2021) call this phenomenon as the *Edge of Stability*.

In Figure 9b, we observe that gradients closely align with dominant subspaces in the EoS regime. This phenomenon stands in contrast with GD in the GF regime (Phenomenon 3), where $\chi_k(\nabla L(\theta_t))$ remains near 0 (see Section D.1 for details).

**Remark 2.** *We highlight that the mechanisms behind gradient alignment differ between SGD in the GF regime and GD in the EoS regime. For SGD in the GF regime, where sharpness stably increases, alignment is due to the stochastic noise and the ill-conditioned loss landscape. In contrast, for GD in the EoS regime, where sharpness oscillates around MSS, alignment arises from a self-stabilization mechanism (Damian et al., 2023), which illustrates that GD oscillates within the dominant subspace.*

Given that the gradient approximately aligns with the dominant subspace, we run experiments analogous to Section 3. Specifically, we train neural networks using GD with a large learning rate $\eta$ until it reaches the EoS regime. We then switch GD to the following update schemes:

$$\theta_{t+1} \leftarrow \theta_t - \eta P_k(\theta_t)\nabla L(\theta_t) , \qquad \text{(Dom-GD)}$$

$$\theta_{t+1} \leftarrow \theta_t - \eta P_k^{\perp}(\theta_t)\nabla L(\theta_t) . \qquad \text{(Bulk-GD)}$$

As shown in Figure 9c, we observe that **Dom-GD fails to further decrease the training loss**, unlike GD. Moreover, **Bulk-GD is as effective as GD in decreasing the training loss**, despite only a small fraction of updates aligning with the bulk subspace.

We provide additional experiments on GD and SGD in the EoS regime in Appendix E. Notably, for SGD in the EoS regime, both stochastic noise and self-stabilization effect affect training dynamics, leading to gradient alignment with dominant subspace, while Dom-SGD still fails to decrease the loss.

## 5.2 SHARPNESS-AWARE MINIMIZATION

Sharpness-Aware Minimization (SAM) (Foret et al., 2021) is a gradient-based optimization method designed to find flat minima. SAM has gained significant attention for its success in practice, especially in improving the generalization performance of deep learning models. For concreteness, we focus on the full-batch version of SAM applied to GD as the base optimizer. This leads to the following update equation:

$$\theta_{t+1} \leftarrow \theta_t - \eta \nabla L \left( \theta_t + \rho \frac{\nabla L(\theta_t)}{\|\nabla L(\theta_t)\|_2} \right) ,$$

where $\eta$ is the learning rate and $\rho$ represents the perturbation radius.

A recent study (Long & Bartlett, 2024) highlights that SAM also operates in its own Edge of Stability regime, wherein the sharpness $\lambda_1(\theta_t)$ saturates near SAM's stability threshold (see Figure 10a). This threshold, denoted as the *SAM-edge*, is defined as:

$$\frac{\|\nabla L(\theta_t)\|_2}{2\rho} \left( \sqrt{1 + \frac{8}{\eta \|\nabla L(\theta_t)\|_2}} - 1 \right) . \qquad \text{(SAM-edge)}$$

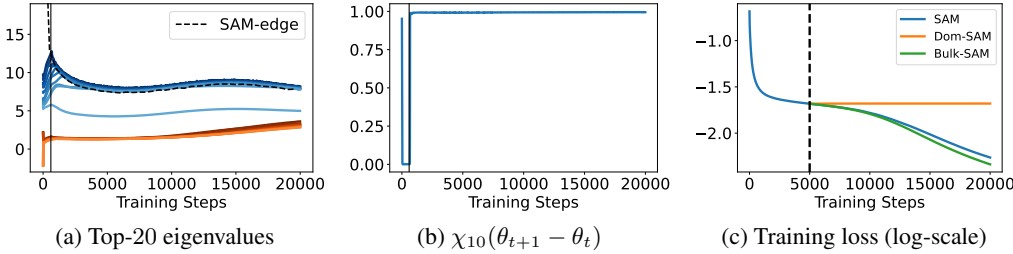

(a) Top-20 eigenvalues
(b) $\chi_{10}(\theta_{t+1} - \theta_t)$
(c) Training loss (log-scale)

Figure 10: **SAM updates approximately lie in dominant subspaces.** Training MLP on MNIST-5k with SAM using a learning rate $\eta = 0.01$ and a perturbation radius $\rho = 0.1$. (a) The plot shows the top-10 eigenvalues in blue and the next top-10 eigenvalues in orange. After a few steps, SAM operates in the EoS regime, where the sharpness stabilizes near the SAM-edge. (b) As the sharpness reaches the SAM-edge, $\chi_{10}(\theta_{t+1} - \theta_t)$ shoots up and remains near 1. (c) We switch the optimizer from SAM to Dom-SAM and Bulk-SAM at step 5000. **Dom-SAM fails to further decrease the training loss, in contrast to SAM and Bulk-SAM.**

Note that GD's stability threshold $2/\eta$ remains constant during training, while SAM-edge is a decreasing function of the norm of the gradient, so it tends to decrease during training.

As shown in Figure 10b, we observe that update vectors of SAM approximately align with dominant subspaces when sharpness saturates near the SAM-edge, similar to GD in the EoS regime. Next, we conduct experiments akin to Section 3: we train neural networks using SAM until the alignment occurs. Then, we switch SAM to the following schemes:

$$\theta_{t+1} \leftarrow \theta_t - \eta P_k(\theta_t)\nabla L\left(\theta_t + \rho\frac{\nabla L(\theta_t)}{\|\nabla L(\theta_t)\|_2}\right),\qquad\text{(Dom-SAM)}$$

$$\theta_{t+1} \leftarrow \theta_t - \eta P_k^\perp(\theta_t)\nabla L\left(\theta_t + \rho\frac{\nabla L(\theta_t)}{\|\nabla L(\theta_t)\|_2}\right).\qquad\text{(Bulk-SAM)}$$

We observe that **Dom-SAM fails to further decrease the training loss, unlike SAM and Bulk-SAM**, as depicted in Figure 10c. Our investigation with various practical algorithms suggests that *neural networks cannot be trained within the dominant subspace, and the bulk subspace plays an important role in learning.*

## 6 MOMENTUM AND ADAPTIVE METHODS AMPLIFY UPDATES IN BULK SUBSPACES

In this section, we further extend our investigation to momentum optimizers, *e.g.*, SGD with momentum, and adaptive optimizers, *e.g.*, Adam. In Appendix F.1, we show that momentum and adaptive learning rates lead to less alignment between the update vector and dominant subspace, so Phenomenon 1 no longer holds for this case. However, Phenomenon 2 still holds, *i.e.*, if we project the update vector onto the dominant subspace (running Dom-SGDM or Dom-Adam), the training loss fails to decrease. This observation also demonstrates that bulk space is where the learning happens.

We build on our results so far and propose explanations for why momentum and adaptive methods are effective for neural network training. At a high level, we claim that they speed up training by amplifying the *bulk component* of each update step. Formally, we introduce the following notion.

**Definition 4** (**Effective learning rate**). *For a given optimization trajectory $\{\theta_t\}$, we define the dominant effective learning rate (Dom-LR) at step $t$ as:*

$$\eta_t^{\text{dom}} := \frac{\langle\theta_t - \theta_{t+1}, P_k(\theta_t)\nabla L(\theta_t)\rangle}{\|P_k(\theta_t)\nabla L(\theta_t)\|_2^2},\qquad\text{(Dom-LR)}$$

*and the bulk effective learning rate (Bulk-LR) at step $t$ as:*

$$\eta_t^{\text{bulk}} := \frac{\langle\theta_t - \theta_{t+1}, P_k^\perp(\theta_t)\nabla L(\theta_t)\rangle}{\|P_k^\perp(\theta_t)\nabla L(\theta_t)\|_2^2}.\qquad\text{(Bulk-LR)}$$

Table 1: Mean effective learning rates over the first 1000 steps (numbers in parentheses show standard deviation). Training Transformer on SST2-1k using GD and Adam with (+m) and without (-m) momentum. GD uses a learning rate of $0.01$, and Adam uses a learning rate of $0.001$. Momentum is set to $\beta = 0.9$.

| Method | Mean Dom-LR | Mean Bulk-LR |
|---|---|---|
| GD(-m) | 0.0100 (0.0000) | **0.0100** (0.0000) |
| GD(+m) | 0.0070 (0.0232) | **0.0828** (0.0576) |
| Adam(-m) | 0.0325 (0.0054) | **0.4672** (0.3555) |
| Adam(+m) | 0.0004 (0.0101) | **2.6639** (1.2480) |

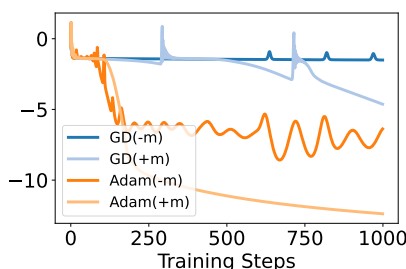

Figure 11: Training loss in log-scale for the experiments in Table 1.

To understand the above notion better, we first consider the simplest case. For a fixed learning rate $\eta$,

- GD has effective learning rates $\eta_t^{\mathrm{dom}} = \eta_t^{\mathrm{bulk}} = \eta$ for all steps $t$.
- Dom-GD has effective learning rates $\eta_t^{\mathrm{dom}} = \eta$ and $\eta_t^{\mathrm{bulk}} = 0$ for all steps $t$.
- Bulk-GD has effective learning rates $\eta_t^{\mathrm{dom}} = 0$ and $\eta_t^{\mathrm{bulk}} = \eta$ for all steps $t$.

Our primary claim in this section is as follows: since Dom-GD fails to decrease the training loss, while Bulk-GD is as effective as GD in reducing the training loss, we claim that Bulk-LR serves as a good indicator for training speed, unlike Dom-LR.

To support this claim, we measure effective learning rates of various optimization methods, including (full-batch) GD with and without momentum, and (full-batch) Adam with and without momentum. Table 1 presents effective learning rates when training Transformer on SST2-1k, and Figure 11 depicts corresponding training loss plot. Additional experiments on other architectures and datasets are provided in Appendix F.2.

Across different settings, we consistently observe that Bulk-LR positively correlates with the training speed. Moreover, momentum and adaptive methods seem to amplify Bulk-LR. This amplification of the bulk component leads to a reduced alignment between the update vector and the dominant subspace, as shown in Appendix F.1. This offers new insights into the effectiveness of momentum and adaptive methods.

## 7 Conclusion and discussion

Motivated by the observation of Gur-Ari et al. (2018) that the gradient aligns with a low-dimensional dominant eigenspace of the training loss Hessian, this work investigates the possiblity of training neural networks within the dominant subspace. Our key contributions are two-fold:

- For every optimizer (e.g., GD, SGD, SGDM, Adam, SAM) we tested, `Dom-OPT`—which projects the update vector onto the dominant subspace—fails to decrease the training loss. This indicates that neural networks *cannot* be trained within the dominant subspace, and bulk subspace plays an essential role during training.
- We identify distinct mechanisms for gradient alignment across different training regimes. In the GF regime, alignment is primarily caused by stochastic noise from SGD in conjunction with the ill-conditioned loss landscape. In the EoS regime, the alignment arises from the self-stabilization mechanism, where oscillations within the dominant subspace lead to this behavior.

**Discussion.** There are remaining questions we leave as a future work. First, providing a theory on our empirical findings beyond the toy model would be an important direction. One interesting observation we did not discuss is that Dom-SGD decreases sharpness and slowly increases the loss, while Bulk-SGD is less noisy and increases sharpness faster than SGD (see Appendix G). Understanding such phenomena would provide deeper insights into neural network training. Lastly, this work focuses on optimization, and exploring the implications to generalization would be important.

ACKNOWLEDGMENTS

This work was partly supported by the National Research Foundation of Korea (NRF) grants funded by the Korean government (MSIT) (No. RS-2023-00211352; No. RS-2024-00421203).

REPRODUCIBILITY STATEMENT

We have made significant efforts to ensure the reproducibility of our results. Detailed experimental settings, including hyperparameters and training details, are provided in Appendix B. Furthermore, to facilitate replication and verification, the source code for the experiments is included in the attached supplementary material. This code contains scripts for reproducing the main results discussed in the paper, along with instructions for running the experiments.

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

# Appendix

## A   RELATED WORK

**Gradient descent in tiny subspaces.**   This work is largely inspired by previous research demonstrating low-rank structures of the training loss Hessian and the gradient in deep neural network training (Sagun et al., 2016; 2017; Gur-Ari et al., 2018). In particular, Jastrzębski et al. (2019) also observe that the SGD update direction is highly aligned with the sharpest direction of the loss landscape. Recently, Schneider et al. (2024) observe that policy gradient algorithms in reinforcement learning also seem to operate in low-dimensional subspaces.

Motivated by such prevalent observations, several follow-up works investigate the possibility of training neural networks in a low-dimensional subspace. If feasible, it has wide applications, including few-shot learning (Gauch et al., 2022) and differential privacy (Singhal & Steinke, 2021; Zhou et al., 2021). Li et al. (2018a) and Gressmann et al. (2020) train neural networks with a small fraction of parameters using random projections, and Li et al. (2023) train ResNet8 on CIFAR10 with a 15-dimensional subspace without sacrificing test accuracy. Note that the results of these work do not contradict our main observation since the low-dimensional subspaces they chose are not the dominant subspace. In particular, Li et al. (2023) construct a low-dimensional subspace by sampling parameter trajectories and then using standard PCA to find the low-dimensional subspace that approximately spans the sampled parameter trajectory. This low-dimensional subspace differs from the dominant subspace we study, as the dominant subspace is defined by the top-$k$ eigenspace of the loss Hessian. Considering our 2D quadratic example (Figure 6) highlights the difference. In this example, our top-1 dominant eigenspace corresponds to the $x$-axis; in contrast, if we consider the PCA of the SGD iterates, the principle component would align much closer to the $y$-axis.

From a theoretical perspective, Arous et al. (2024) rigorously prove that the SGD update aligns with the dominant subspace in multi-class logistic regression and XOR classification with a two-layer network. More recently, Yaras et al. (2024) theoretically prove that for deep overparameterized low-rank matrix recovery, the learning dynamics of each weight matrix are confined to an invariant low-dimensional subspace. However, they consider scenarios where GD in the GF regime align with the low-rank subspace, which is not the case in our settings.

**Edge of Stability.**   Most analyses of GD have focused on settings where the learning rate is sufficiently small to ensure that the training loss monotonically decreases in the GF regime. However, recent empirical studies (Jastrzębski et al., 2019; 2020) observe that GD with practically large learning rates decreases the loss non-monotonically and finds flatter minima. Cohen et al. (2021) call this the *Edge of Stability* (EoS) phenomenon. Subsequent theoretical works have made progress towards understanding the mechanisms of EoS (Ahn et al., 2022; Damian et al., 2023; Wu et al., 2023). Moreover, several recent works theoretically analyze precise training dynamics under simplified models (Ahn et al., 2023; Kreisler et al., 2023; Song & Yun, 2023; Zhu et al., 2023). The self-stabilization mechanism (Damian et al., 2023) shows that GD in the EoS regime exhibits oscillation along the top eigenvector, while the overall decrease in loss occurs due to movement in directions orthogonal to the top eigenvector. Moreover, SGD with large learning rates also operates at the (stochastic) Edge of Stability (Lee & Jang, 2023; Agarwala & Pennington, 2024). Notably, Zhu et al. (2024) observe that catapults in SGD occurs in the low-rank top eigenspace of NTK, which is closely related to the gradient alignment with the dominant subspace.

**Sharpness-Aware Minimization.**   Inspired by prior works (Keskar et al., 2017; Jiang et al., 2020) showing that flat minima often lead to better generalization, Foret et al. (2021) propose an optimization method called *Sharpness-Aware Minimization* (SAM), designed to find flat minima. SAM has shown great success in practice, and subsequently, several works theoretically investigate the dynamics of SAM and its convergence properties (Andriushchenko & Flammarion, 2022; Bartlett et al., 2023; Dai et al., 2023; Si & Yun, 2023; Wen et al., 2023). Recently, Agarwala & Dauphin (2023) and Long & Bartlett (2024) empirically observe that SAM also goes through unstable dynamics, akin to EoS.

**The role of momentum and Adam.**   Momentum (Polyak, 1964; Nesterov et al., 2018) and adaptive methods (Kingma & Ba, 2014) are workhorses for training deep neural network models. Adaptive methods, such as Adam, have gained renewed interest due to their success in training language models (Zhang et al., 2020). However, the current understanding of their effectiveness for neural network training remains incomplete.

The role of momentum is quite well understood for convex settings, through acceleration mechanism (Nesterov et al., 2018; Kidambi et al., 2018). For nonconvex settings, the provable benefits of momentum are investigated for variants of SGD, such as normalized SGD (Cutkosky & Mehta, 2020) and signSGD (Crawshaw et al., 2022). A recent work by Wang et al. (2024) shows that the benefit of momentum is marginal when the learning rate is small and gradient noise is dominant. Moreover, Fu et al. (2023) empirically demonstrate the benefits of momentum for large learning rates from a sharpness perspective.

Adam has been observed to be particularly effective in training transformers (Zhang et al., 2024), even for simplified shallow linear transformers trained on linear regression tasks (Ahn et al., 2024a). Its superiority over SGD has been attributed to factors such as heavy-tailed class imbalances in language tasks (Kunstner et al., 2024) and block heterogeneity in Hessian (Zhang et al., 2024). A recent line of work shows that full-batch Adam is a smoothed version of SignGD (Kunstner et al., 2023; Xie & Li, 2024). Additionally, Ahn et al. (2024c) and Ahn & Cutkosky (2024) study the benefits of Adam from an online learning perspective.

## B EXPERIMENTAL DETAILS

In this section, we provide the details of our experiments which are not covered in the main text.

### B.1 ARCHITECTURES

The main experiments are conducted on three types of architectures: MLP, CNN, and Transformer. Additional experiments conducted on standard architectures are provided in Section C.2.

- **MLP**: We use a 3-layer MLP with a width of 200 and $\tanh$ activation functions, following the architecture used in Cohen et al. (2021).
- **CNN**: We use a 3-layer CNN with a width of 32 and ReLU activation functions, also based on the architecture from Cohen et al. (2021).
- **Transformer**: We use a 2-layer Transformer with a hidden dimension of $64$ and $8$ attention heads, based on the architecture used in Damian et al. (2023).

### B.2 DATA

The main experiments are conducted on three datasets: MNIST-5k, CIFAR10-5k, and SST2-1k. The primary task is classification with categorical MSE loss, and additional experiments with cross-entropy loss are provided in Section C.1.

- **MNIST-5k**: We use the first 5000 samples of MNIST dataset (LeCun et al., 1998) for multi-class classification. The number of classes is 10.
- **CIFAR10-5k**: We use the first 5000 samples of CIFAR10 dataset (Krizhevsky, 2009) for multi-class classification. The number of classes is 10.
- **SST2-1k**: We use the first 1000 samples of SST2 dataset (Socher et al., 2013) for binary classification.

### B.3 EXPERIMENTAL SETUP

Throughout this paper, all experiments are conducted using a constant learning rate. For experiments using SGD, we use a batch size of 50 for all experiments. Below, we provide details on the choice of learning rates for each experiment, which are not specified in the main text.

- Figure 1, Figure 2, Figure 3, Figure 18, Figure 19, and Figure 20: The training loss, eigenvalues of the loss Hessian, and $\chi_k(\nabla L(\theta_t))$ are computed on the same run of SGD/GD with small learning rates. The learning rates used are:
    - MLP on MNIST-5k: 0.01,
    - CNN on CIFAR10-5k: 0.001,
    - Transformer on SST2-1k: 0.001.
- Figure 29, Figure 30, Figure 31, Figure 32, Figure 33, and Figure 34: The training loss, eigenvalues of the loss Hessian, and $\chi_k(\nabla L(\theta_t))$ are computed on the same run of (S)GD with large learning rates. The learning rates used are:
    - MLP on MNIST-5k: 0.1,
    - CNN on CIFAR10-5k: 0.01,
    - Transformer on SST2-1k: 0.005.
- Figure 4 and Figure 5: We train MLP on MNIST-5k using (S)GD with a learning rate of 0.01, under the same initialization.
- Figure 12, Figure 13, and Figure 14: The eigenvalues of the loss Hessian, $\chi_k(\nabla L(\theta_t))$, and the training loss are computed on the same run of SGD. The learning rates used are:
    - MLP on MNIST-5k: 0.1,
    - CNN on CIFAR10-5k: 0.001,
    - Transformer on SST2-1k: 0.001.

- Figure 15, Figure 16, and Figure 17: The eigenvalues of the loss Hessian, $\chi_k(\nabla L(\theta_t))$, and the training loss are computed on the same run of SGD. The learning rates used are:
    - VGG11 on CIFAR10-5k: 0.01,
    - ResNet8 on CIFAR10-5k: 0.01.

Our experiments were conducted using Pytorch (Paszke et al., 2019), and we referred to the GitHub repository at `https://github.com/locuslab/edge-of-stability` to replicate the experimental setup described in Cohen et al. (2021). All experiments were performed on a single server equipped with $4$ NVIDIA RTX 3090 GPUs.

# C ADDITIONAL EXPERIMENTS FOR SECTION 3

In this section, we provide additional experimental results to support the observations made in Section 3. These experiments demonstrate that our critical observation—that Dom-SGD fails to further decrease the training loss—also holds when using cross-entropy loss and training with standard architectures.

## C.1 CROSS-ENTROPY LOSS

We use cross-entropy loss instead of MSE loss for classification tasks, and provide the results analgous to Figure 1, Figure 2, and Figure 3.

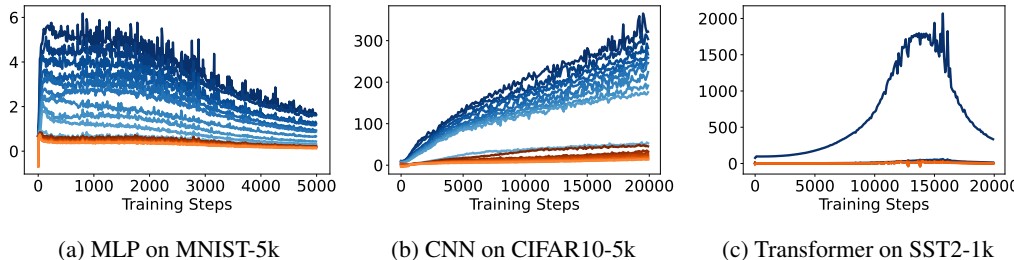

(a) MLP on MNIST-5k      (b) CNN on CIFAR10-5k      (c) Transformer on SST2-1k

Figure 12: **Low-rank structure of the Hessian.** The plot shows the top eigenvalues of the loss Hessian during SGD training. The blue curves represent the top-$k$ eigenvalues, which are significantly larger than the next top-$k$ eigenvalues, shown in orange, where $k$ is the number of classes for the classification task.

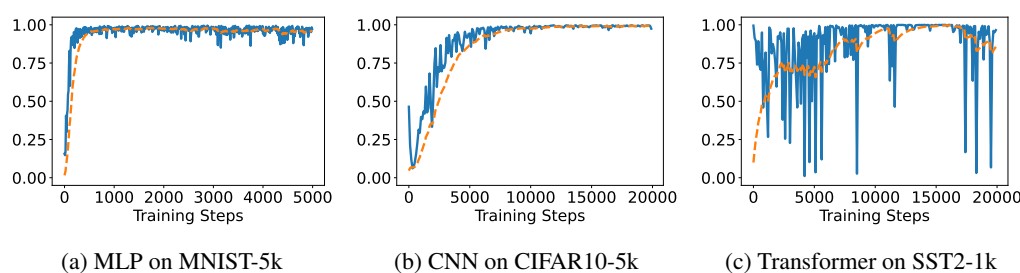

(a) MLP on MNIST-5k      (b) CNN on CIFAR10-5k      (c) Transformer on SST2-1k

Figure 13: **Alignment of gradients with dominant subspaces.** The plot illustrates $\chi_k(\nabla L(\theta_t))$ during SGD training. The orange dashed lines represent the exponential moving average (EMA) of $\chi_k(\nabla L(\theta_t))$. After a few early steps, $\chi_k(\nabla L(\theta_t))$ reaches and stays near 1, indicating the alignment between gradients and dominant subspaces.

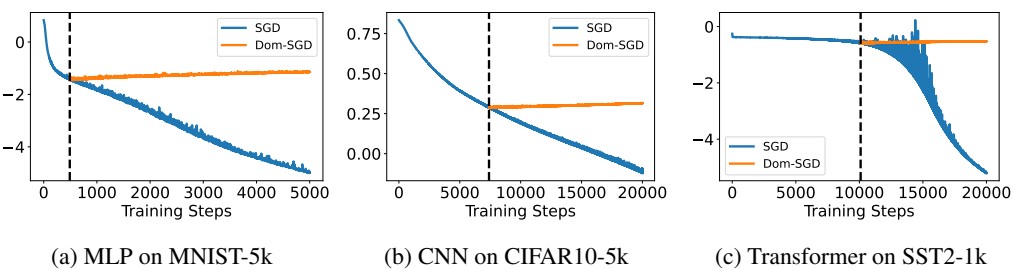

(a) MLP on MNIST-5k      (b) CNN on CIFAR10-5k      (c) Transformer on SST2-1k

Figure 14: Training loss (log-scale) of SGD and Dom-SGD. **Dom-SGD fails to further decrease the training loss** in contrast to SGD, despite the gradients aligning approximately with the dominant subspace. We switch from SGD to Dom-SGD whenever the EMA value of $\chi_k(\nabla L(\theta_t))$ exceeds 0.95.

## C.2 STANDARD ARCHITECTURES ON CIFAR10-5K

To ensure the generality of our observations, we conducted experiments on standard architectures, specifically VGG11 and ResNet8, using CIFAR10-5k dataset, using MSE loss.

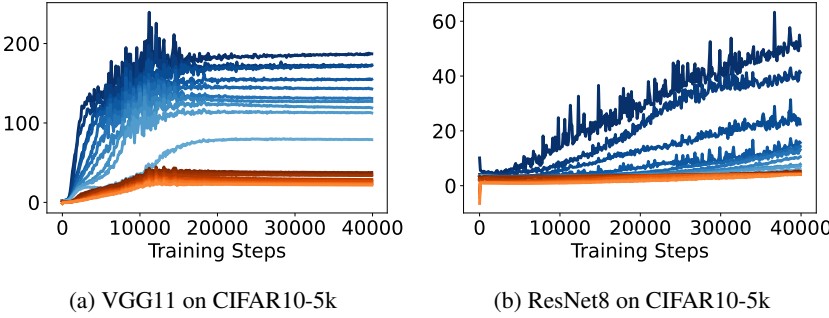

(a) VGG11 on CIFAR10-5k  (b) ResNet8 on CIFAR10-5k

Figure 15: **Low-rank structure of the Hessian.** The plot shows the top eigenvalues of the loss Hessian during SGD training. The blue curves represent the top-10 eigenvalues, which are significantly larger than the next top-10 eigenvalues, shown in orange.

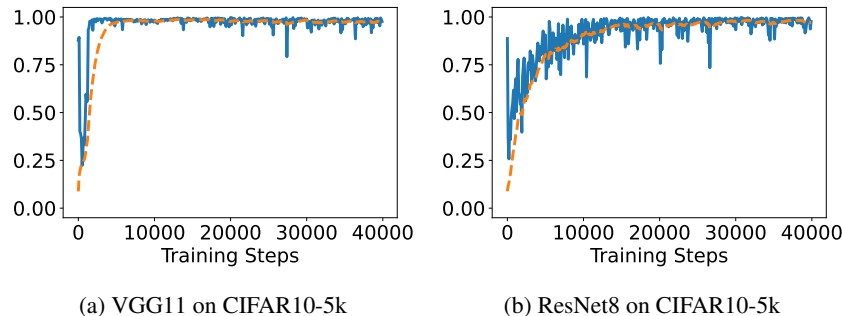

(a) VGG11 on CIFAR10-5k  (b) ResNet8 on CIFAR10-5k

Figure 16: **Alignment of gradients with dominant subspaces.** The plot illustrates $\chi_{10}(\nabla L(\theta_t))$ during SGD training. The orange dashed lines represent the exponential moving average (EMA) of $\chi_{10}(\nabla L(\theta_t))$. After a few early steps, $\chi_{10}(\nabla L(\theta_t))$ reaches and stays near 1, indicating the alignment between gradients and dominant subspaces.

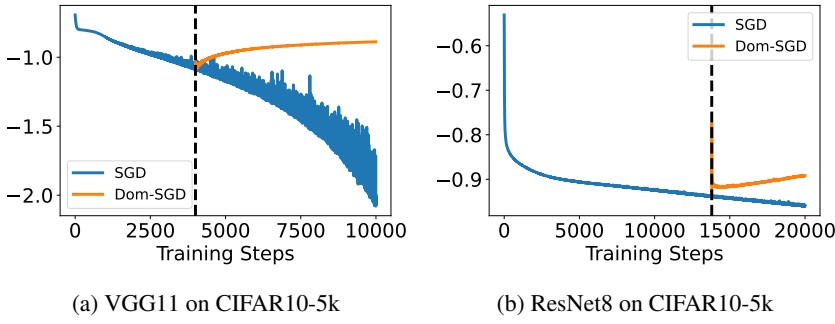

(a) VGG11 on CIFAR10-5k  (b) ResNet8 on CIFAR10-5k

Figure 17: Training loss (log-scale) of SGD and Dom-SGD. **Dom-SGD fails to further decrease the training loss** in contrast to SGD, despite the gradients aligning approximately with the dominant subspace. We switch from SGD to Dom-SGD whenever the EMA value of $\chi_k(\nabla L(\theta_t))$ exceeds 0.95.

# D    ADDITIONAL EXPERIMENTS FOR SECTION 4

This section provides additional experimental results to support the observations made in Section 4.

## D.1    NO ALIGNMENT WITH DOMINANT SUBSPACES ALONG GD TRAJECTORIES

We run GD in the GF regime under the same settings as Figure 3, using the same learning rate and initialization. Figure 18 shows the top Hessian eigenvalues during training. The smooth increase of the Hessian eigenvalues indicate that the training is happening at the *GF regime*. The corresponding gradient alignment is shown in Figure 19, and we observe that $\chi_k(\nabla L(\theta_t))$ quickly approaches and remains near $0$, indicating that gradients do not align with dominant subspaces, unlike in SGD in the GF regime. Figure 20 shows that Dom-GD fails to further decrease the training loss in this scenario.

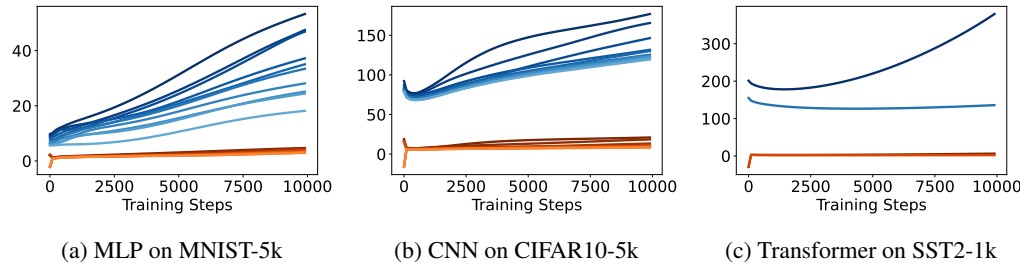

(a) MLP on MNIST-5k      (b) CNN on CIFAR10-5k      (c) Transformer on SST2-1k

Figure 18: The plot shows the top eigenvalues of the loss Hessian during GD training in the GF regime. The blue curves represent the top-$k$ eigenvalues, which are significantly larger than the next top-$k$ eigenvalues, shown in orange, where $k$ is the number of classes for the classification task.

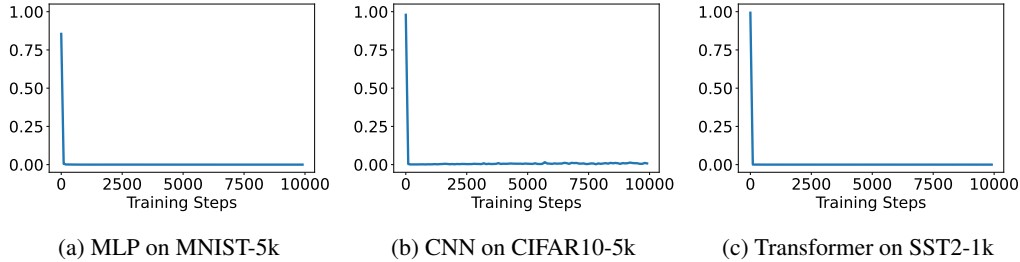

(a) MLP on MNIST-5k      (b) CNN on CIFAR10-5k      (c) Transformer on SST2-1k

Figure 19: **Gradients do not align with dominant subspaces on GD trajectories.** The plot illustrates $\chi_k(\nabla L(\theta_t))$ during GD training in the GF regime. After a few early steps, $\chi_k(\nabla L(\theta_t))$ reaches and stays near $0$, indicating alignment with bulk subspaces. The same learning rates and initializations as Figure 3 are used.

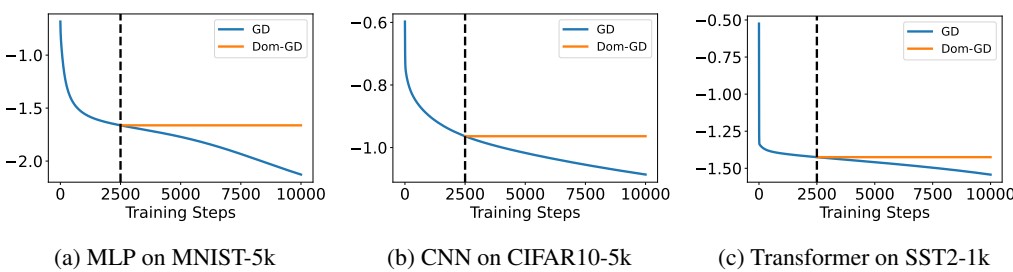

(a) MLP on MNIST-5k      (b) CNN on CIFAR10-5k      (c) Transformer on SST2-1k

Figure 20: Training loss (log-scale) of Dom-GD and GD. Dom-GD fails to further decrease the training loss, unlike GD. We switch from GD to Dom-GD at iteration $2500$.

## D.2 SGD TRAJECTORIES TRACK GRADIENT FLOW

Figure 3 and Figure 19 suggest that GD and SGD exhibit different alignments with dominant subspaces, even when using the same learning rate and initialization. However, both should track the continuous-time gradient flow trajectory, if the learning rate is sufficiently small. In Figure 21, we confirm that the trajectory of GD $\{\theta_t^{\mathrm{GD}}\}$ and the trajectory of SGD $\{\theta_t^{\mathrm{SGD}}\}$ are close to each other. Specifically, we observe that $\left\|\theta_t^{\mathrm{GD}} - \theta_t^{\mathrm{SGD}}\right\| \ll \left\|\theta_t^{\mathrm{GD}} - \theta_0^{\mathrm{GD}}\right\| \approx \left\|\theta_t^{\mathrm{SGD}} - \theta_0^{\mathrm{SGD}}\right\|$, indicating the trajectories are quite similar.

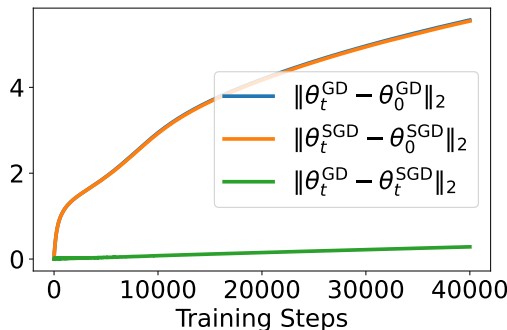

Figure 21: Trajectories of GD and SGD are close to each other.

## D.3 SWITCHING BETWEEN GD AND SGD

Here, we provide additional plots for the experiments shown in Figure 4 and Figure 5. Figure 22 illustrates $\chi_k(\nabla L(\theta_t))$, training loss, and top eigenvalues of the loss Hessian when switching from SGD to GD, corresponding to the experiment in Figure 4. Figure 23 illustrates $\chi_k(\nabla L(\theta_t))$, training loss, and top eigenvalues of the loss Hessian when switching from GD to SGD, corresponding to the experiment in Figure 5.

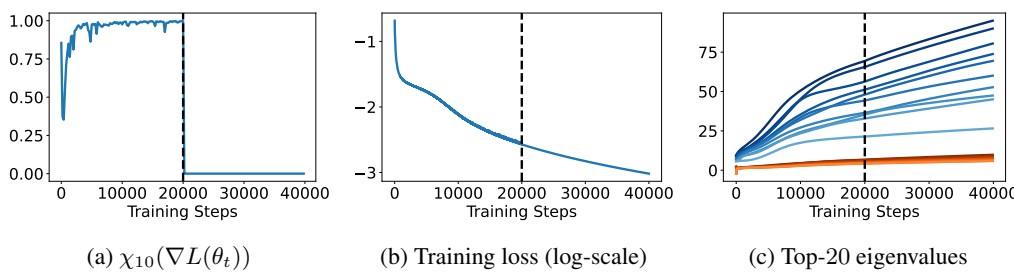

(a) $\chi_{10}(\nabla L(\theta_t))$       (b) Training loss (log-scale)       (c) Top-20 eigenvalues

Figure 22: The left plot shows $\chi_k(\nabla L(\theta_t))$ when training MLP on MNIST-5k. A sharp transition in gradient alignment with the dominant subspace is observed when switching from SGD to GD (same plot as Figure 4). The plots of the training loss and top eigenvalues of the loss Hessian change relatively smoothly when switching from SGD to GD, in contrast to $\chi_{10}(\nabla L(\theta_t))$. In the right plot, the blue curves represent the top-10 eigenvalues, and the orange curves represent the next top-10 eigenvalues.

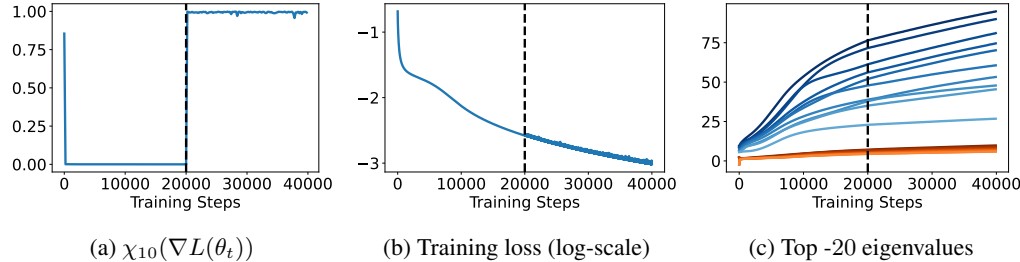

(a) $\chi_{10}(\nabla L(\theta_t))$          (b) Training loss (log-scale)          (c) Top -20 eigenvalues

Figure 23: The left plot shows $\chi_k(\nabla L(\theta_t))$ when training MLP on MNIST-5k. A sharp transition in gradient alignment with the dominant subspace is observed when switching from GD to SGD (same plot as Figure 5). The plots of the training loss and top eigenvalues of the loss Hessian change relatively smoothly when switching from GD to SGD, in contrast to $\chi_{10}(\nabla L(\theta_t))$. In the right plot, the blue curves represent the top-10 eigenvalues, and the orange curves represent the next top-10 eigenvalues.

### D.4    EFFECT OF NOISE SCALE ON SPURIOUS ALIGNMENT

We train MLP on MNIST-5k using Noisy Gradient Descent (NGD) with varying noise scales. NGD is implemented by injecting noise sampled from an isotropic Gaussian distribution after each GD update iteration. We use the same learning rate and initialization as Figure 19. We observe that higher noise scales increase the alignment between the gradient and the dominant subspace.

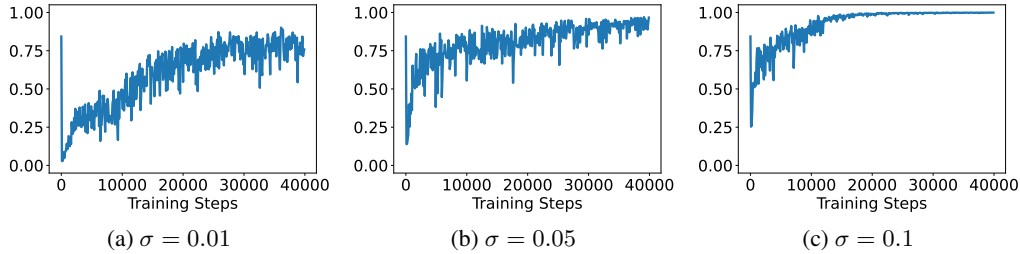

(a) $\sigma = 0.01$          (b) $\sigma = 0.05$          (c) $\sigma = 0.1$

Figure 24: **Effect of noise scale in NGD on alignment of gradients with dominant subspace.** The plot illustrates $\chi_{10}(\nabla L(\theta_t))$ during Noisy Gradient Descent (NGD) training MLP on MNIST-5k with varying noise scales $\sigma$. NGD is implemented by injecting noise from a Gaussian distribution $N(0, \eta^2\sigma^2 I)$ after each GD update iteration. We observe that higher noise scales increase the alignment between gradient and dominant subspace, supporting our finding that noise causes spurious alignment in SGD.

We train MLP on MNIST-5k using SGD with varying batch sizes by $\{50, 100, 500, 1000, 5000\}$ under the same initialization and learning rate. We consider training in the GF regime where GD does not exhibit alignment, using the same learning rate as Figure 19. We observe that smaller batch sizes (i.e., higher noise scale) increase the alignment between the gradient and the dominant subspace. Moreover, the loss curves are similar to each other, indicating that the trajectories are closely tracking the continuous-time gradient flow.

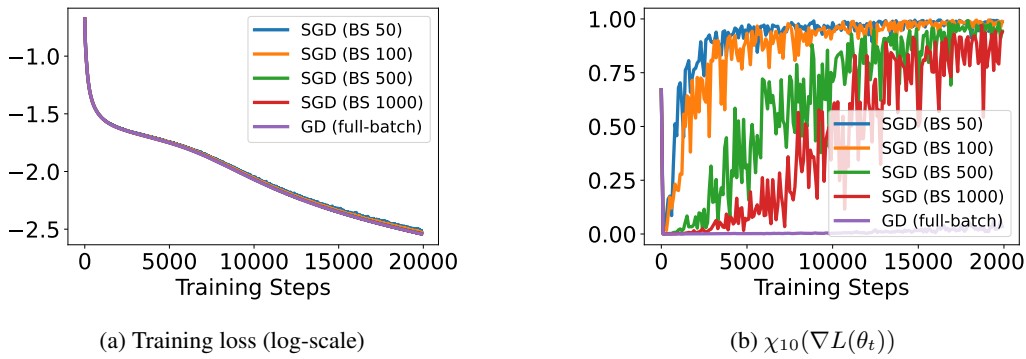

(a) Training loss (log-scale)

(b) $\chi_{10}(\nabla L(\theta_t))$

Figure 25: **Effect of batch size in SGD on alignment of gradients with dominant subspace.** The plot illustrates $\chi_{10}(\nabla L(\theta_t))$ during SGD training MLP on MNIST-5k with varying batch sizes by BS $\in \{50, 100, 500, 1000, 5000\}$. We observe that smaller batch sizes increase the alignment between gradient and dominant subspace, supporting our finding that noise causes spurious alignment in SGD.

### D.5 DISTANCE FROM "SWITCHING" STEP

We measure the $\ell_2$ distance of the weights from the step where we switch from SGD to Dom-SGD and Bulk-SGD for the experiments in Figure 1. As shown in Figure 26, the distance from the switching step remains relatively small and tends to saturate for Dom-SGD. In contrast, for both SGD and Bulk-SGD, the distance from the switching step increases much faster and in a similar manner. This observation is consistent with the explanation based on an ill-conditioned-valley-like landscape proposed in Section 4.3.

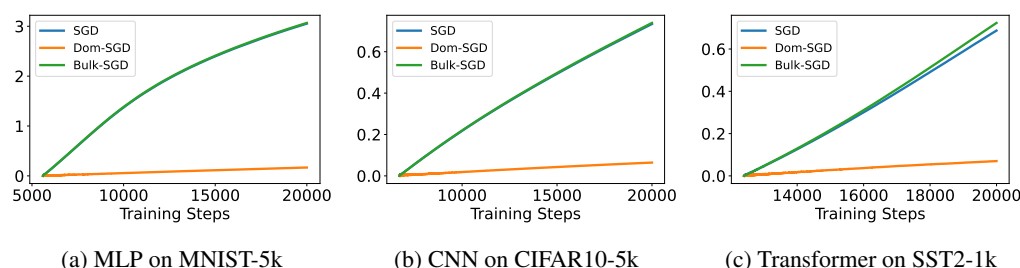

(a) MLP on MNIST-5k

(b) CNN on CIFAR10-5k

(c) Transformer on SST2-1k

Figure 26: The plots illustrate the $\ell_2$ distance of the weights from the step where we switch from SGD to Dom-SGD and Bulk-SGD for the experiments in Figure 1. We observe that Dom-SGD fails to make further progress in terms of distance, in contrast to SGD and Bulk-SGD.

We also measure the $\ell_2$ distance of the weights from the step where we switch from GD to Dom-GD for the experiments in Figure 20. As seen in Figure 27, the distance remains near zero for Dom-GD, indicating minimal movement. While Dom-SGD exhibits slight movement due to its stochastic nature, Dom-GD shows near-zero movement, suggesting it gets stuck at the bottom of the valley, consistent with the ill-conditioned-valley-like landscape described in Section 4.3.

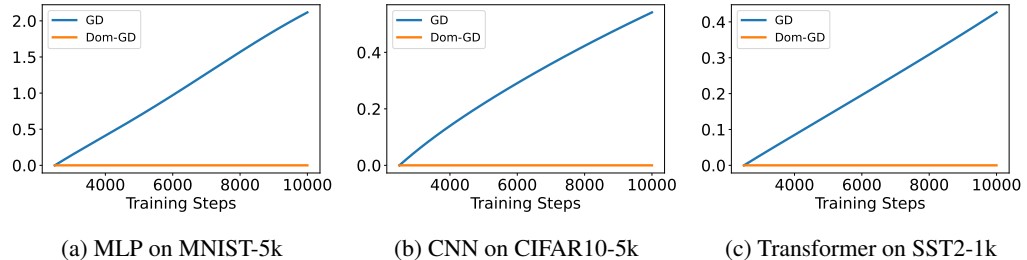

(a) MLP on MNIST-5k    (b) CNN on CIFAR10-5k    (c) Transformer on SST2-1k

Figure 27: The plots illustrate the $\ell_2$ distance of the weights from the step where we switch from GD to Dom-GD for the experiments in Figure 20. We observe that Dom-GD gets stuck at the initial point.

### D.6    TOY EXAMPLE

We conduct additional experiments using Noisy GD for a toy model we considered in Section 4.2. We consider the same 2-dimensional ill-conditioned quadratic loss, $L(x, y) = \frac{1}{2}(1000x^2 + y^2)$, where $\theta = (x, y) \in \mathbb{R}^2$. We run GD with learning rate $\eta$ as

$$\theta_{t+1}^{\mathrm{GD}} \leftarrow \theta_t^{\mathrm{GD}} - \eta \nabla L(\theta_t^{\mathrm{GD}}),$$

and Noisy GD implemented by injecting a Gaussian noise after each GD update

$$\theta_{t+1}^{\mathrm{NGD}} \leftarrow \theta_t^{\mathrm{NGD}} - \eta \nabla L(\theta_t^{\mathrm{NGD}}) + \eta \epsilon_t, \quad \text{where } \epsilon_t \sim \mathcal{N}(0, \sigma^2 I).$$

In Figure 28, we visualize the optimization trajectories of GD and Noisy GD with an initialization $\theta_0^{\mathrm{GD}} = \theta_0^{\mathrm{NGD}} = (1, 1)$, learning rate $\eta = 10^{-4}$, and noise scale $\sigma^2 = 10$. We also compute the fraction of gradient in the dominant subspace as $\chi_1(\nabla L(\theta)) := \frac{|\langle \nabla L(\theta), e_1 \rangle|}{\|\nabla L(\theta)\|_2}$. In both GD and Noisy GD trajectories, $x_t$ quickly converges to $0$, and both trajectories remain close to the $y$-axis throughout the remaining of the training. However, in GD, $\chi_1(\nabla L(\theta_t^{\mathrm{GD}}))$ quickly approaches and remains near $0$, while in Noisy GD, $\chi_1(\nabla L(\theta_t^{\mathrm{NGD}}))$ stays close to 1.

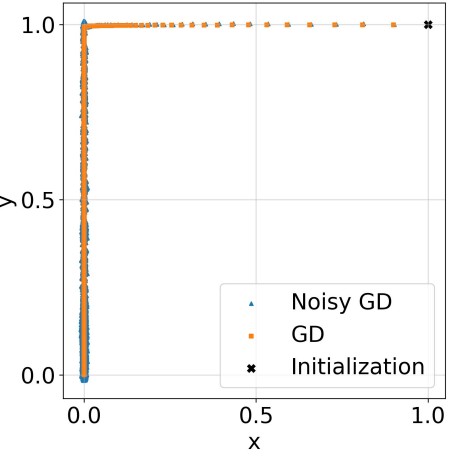
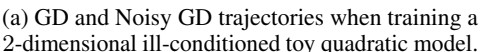

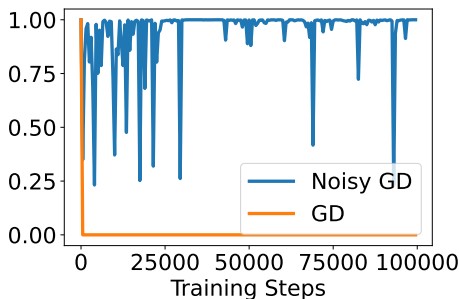

(a) GD and Noisy GD trajectories when training a 2-dimensional ill-conditioned toy quadratic model.

(b) $\chi_1(\nabla L(\theta_t))$ during GD and Noisy GD

Figure 28: We train an ill-conditioned quadratic loss $L(x, y) = \frac{1}{2}(1000x^2 + y^2)$ using GD and Noisy GD. Note that spurious alignment is observed for Noisy GD, unlike GD.

## E  SPURIOUS ALIGNMENT IN THE EoS REGIME

This section provides additional experimental results on GD and SGD in the EoS regime to support our observations made in Section 5.1.

Figure 29 shows the evolution of top Hessian eigenvalues during training. The oscillations of the sharpness indicate that the training is happening at the EoS regime. The corresponding gradient alignment is shown in Figure 30, and we observe that $\chi_k(\nabla L(\theta_t))$ remains near 1 at the EoS regime, indicating that gradients align with the dominant subspace, unlike in GD in the GF regime. Figure 20 shows that Dom-GD fails to further decrease the training loss, despite gradients aligning on the dominant subspace.

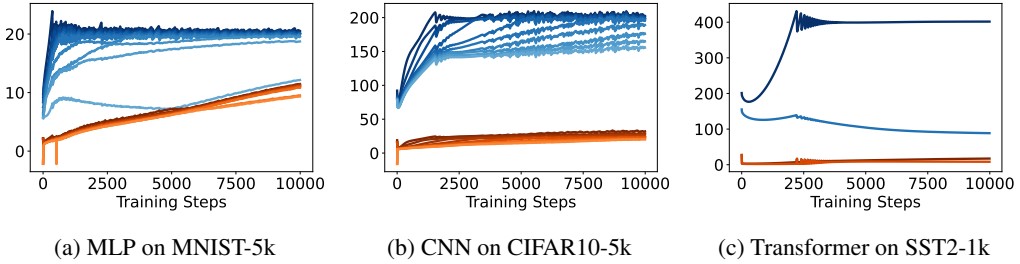

|  (a) MLP on MNIST-5k | (b) CNN on CIFAR10-5k | (c) Transformer on SST2-1k |

Figure 29: The plot shows the top eigenvalues of the loss Hessian during GD training with large learning rates. The blue curves represent the top-$k$ eigenvalues, which are significantly larger than the next top-$k$ eigenvalues, shown in orange. After a few initial steps, GD enters the EoS regime, where the sharpness stabilizes near $2/\eta$.

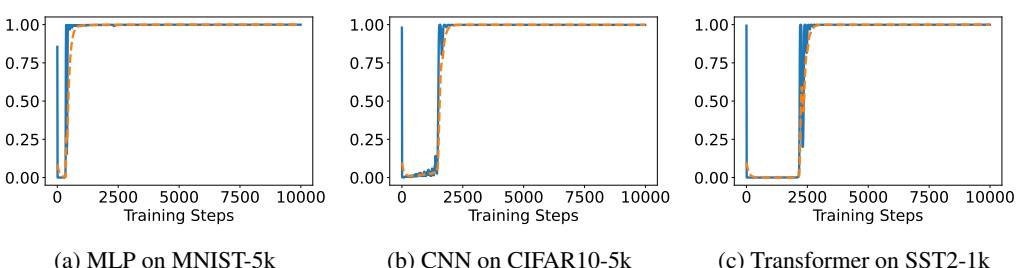

|  (a) MLP on MNIST-5k | (b) CNN on CIFAR10-5k | (c) Transformer on SST2-1k |

Figure 30: **Gradients approximately align with dominant subspaces in the EoS regime.** The plot illustrates $\chi_k(\nabla L(\theta_t))$ during GD training with large learning rates. The orange dashed lines represent the exponential moving average (EMA) of $\chi_{10}(\nabla L(\theta_t))$. As the sharpness reaches $2/\eta$, $\chi_{10}(\nabla L(\theta_t))$ approaches and remains near 1, indicating the gradient alignment with dominant subspace.

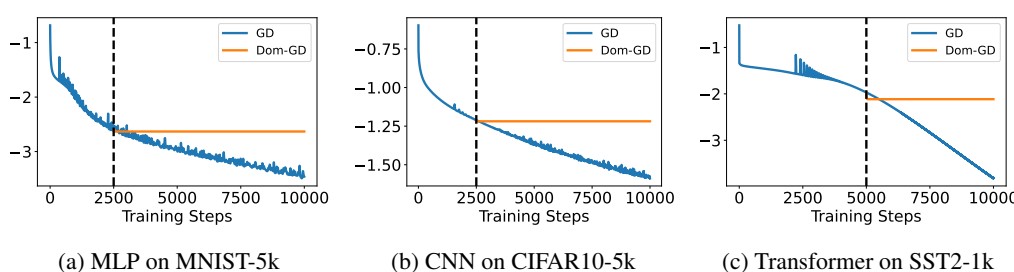

|  (a) MLP on MNIST-5k | (b) CNN on CIFAR10-5k | (c) Transformer on SST2-1k |

Figure 31: Training loss (log-scale) of GD and Dom-GD. **Dom-GD fails to further decrease the training loss** in contrast to GD, despite the gradients aligning with the dominant subspace. We switch from GD to Dom-GD after GD reaches the EoS regime where $\chi_k(\nabla L(\theta_t))$ stays near 1.

Similarly, the same set of experiments on SGD with large learning rates are shown in Figures 32, 33, and 34. In this scenario, both stochastic noise and self-stabilization effect affects the training dynamics. We observe the gradient alignment in this scenario, and Dom-SGD again fails to further decrease the training loss. Interestingly, Dom-SGD rather increases the training loss.

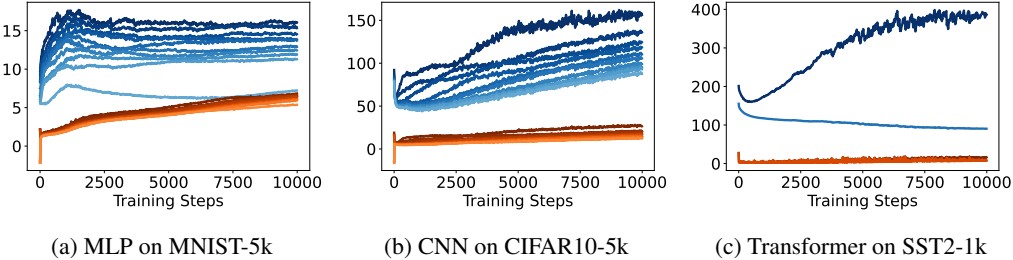

(a) MLP on MNIST-5k      (b) CNN on CIFAR10-5k      (c) Transformer on SST2-1k

Figure 32: The plot shows the top eigenvalues of the loss Hessian during SGD training with large learning rates. The blue curves represent the top-$k$ eigenvalues, which are significantly larger than the next top-$k$ eigenvalues, shown in orange.

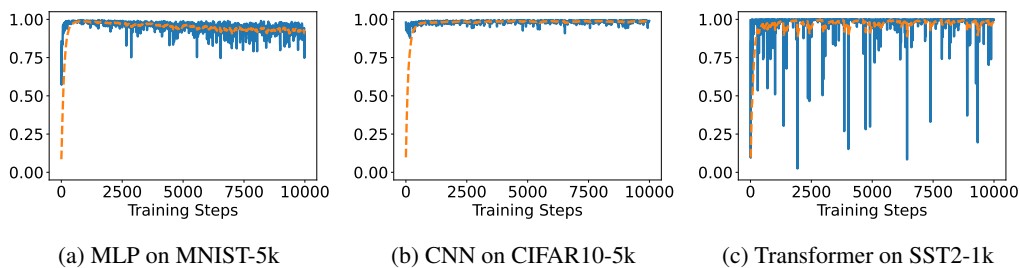

(a) MLP on MNIST-5k      (b) CNN on CIFAR10-5k      (c) Transformer on SST2-1k

Figure 33: The plot illustrates $\chi_k(\nabla L(\theta_t))$ during SGD training with large learning rates. The orange dashed lines represent the exponential moving average (EMA) of $\chi_{10}(\nabla L(\theta_t))$. After a few initial steps, $\chi_{10}(\nabla L(\theta_t))$ approaches and remains near 1, indicating the gradient alignment with dominant subspace.

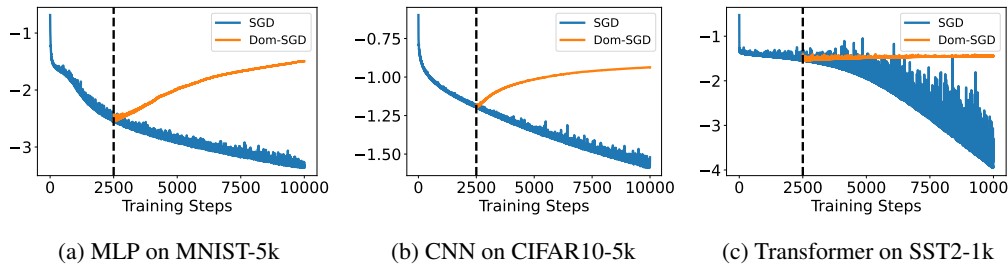

(a) MLP on MNIST-5k      (b) CNN on CIFAR10-5k      (c) Transformer on SST2-1k

Figure 34: Training loss (log-scale) of SGD and Dom-SGD. **Dom-SGD fails to further decrease the training loss** in contrast to SGD, despite the gradients aligning approximately with the dominant subspace. We switch from SGD to Dom-SGD after $\chi_k(\nabla L(\theta_t))$ stabilizes near 1.

## F  ADDITIONAL EXPERIMENTS FOR SECTION 6

This section provides additional experimental results to support the observations made in Section 6.

### F.1  SGDM AND ADAM

We train MLP on MNIST-5k using SGD with momentum (SGDM) and Adam. The results are shown in Figure 35 and Figure 36. We observe that SGDM and Adam updates are "partially" aligned with the dominant subspace due to the effects of momentum and adaptive learning rates. Interestingly, the use of momentum and adaptive methods leads to less alignment between the gradient and dominant subspace than SGD, consistent with the observations on effective learning rates Section 6. We also implement Dom-SGDM and Dom-Adam, which project each SGDM/Adam update onto the dominant subspace. We observe that Dom-SGDM and Dom-Adam fail to further decrease the training loss, unlike SGDM and Adam, indicating that the update component aligned with the dominant subspace does not contribute to the loss decrease.

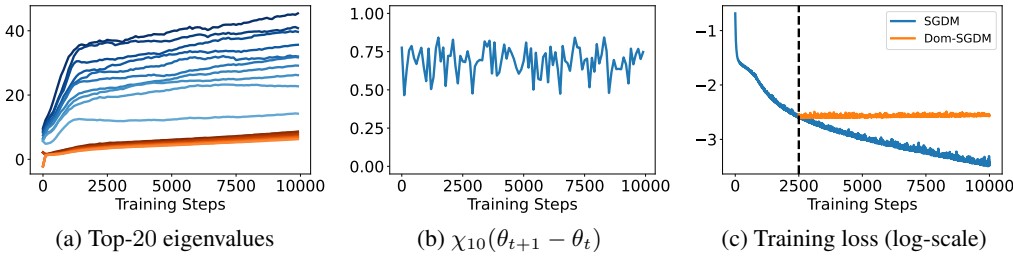

    (a) Top-20 eigenvalues         (b) $\chi_{10}(\theta_{t+1} - \theta_t)$         (c) Training loss (log-scale)

Figure 35: **SGDM updates partially align with the dominant subspace.** Training MLP on MNIST-5k with SGDM using a learning rate $\eta = 0.01$ and a momentum factor $\beta = 0.9$. (a) Top-10 eigenvalues (blue curves) are significantly larger than the next top-10 eigenvalues (orange curves). (b) $\chi_{10}(\theta_{t+1} - \theta_t)$ values remain in the range of $(0.5, 0.8)$, indicating that SGDM updates are partially aligned with the dominant subspace due to the effect of momentum. (c) Switching the optimizer from SGDM to Dom-SGDM at step 2500 shows that **Dom-SGDM fails to further decrease the training loss, unlike SGDM**.

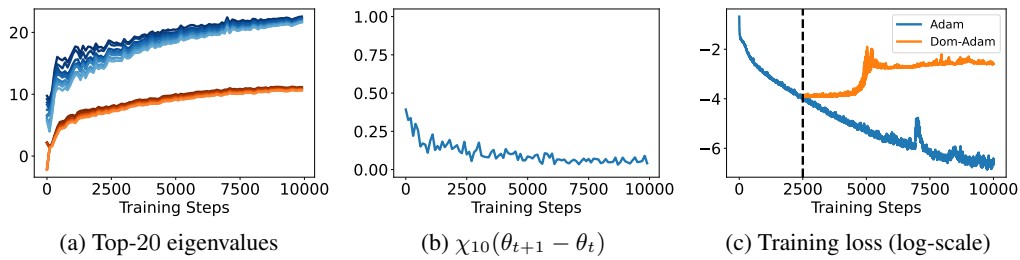

    (a) Top-20 eigenvalues         (b) $\chi_{10}(\theta_{t+1} - \theta_t)$         (c) Training loss (log-scale)

Figure 36: **Adam updates partially align with the dominant subspace.** Training MLP on MNIST-5k with Adam using a learning rate $\eta = 0.001$ and momentum factors $(\beta_1, \beta_2) = (0.9, 0.999)$. (a) Top-10 eigenvalues (blue curves) are significantly larger than the next top-10 eigenvalues (orange curves). (b) $\chi_{10}(\theta_{t+1} - \theta_t)$ values remain in the range of $(0.05, 0.5)$, indicating that Adam updates are partially aligned with the dominant subspace. Interestingly, the alignment tends to decrease during training due to the effect of adaptive learning rates. (c) Switching the optimizer from Adam to Dom-Adam at step 2500 shows that **Dom-Adam fails to further decrease the training loss, unlike Adam**.

### F.2  EFFECTIVE LEARNING RATES

In Figure 37, we plot effective learning rates (smoothed with EMA) as a function over time for experiments in Figure 1.

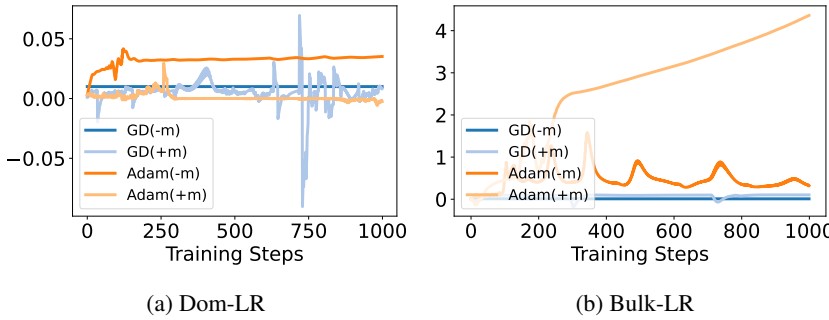

(a) Dom-LR  (b) Bulk-LR

Figure 37: Effective learning rates (smoothed with EMA) for experiments in Figure 1 when training Transformer on SST2-1k.

We measure the effective learning rates for (full-batch) GD with and without momentum, and (full-batch) Adam with and without momentum on different architectures and datasets. Tables 2 and 3 present the effective learning rates when training an MLP on MNIST-5k and a CNN on CIFAR10-5k. Figures 38 and 40 show the corresponding training loss plots.

Figure 39 and Figure 41 show the corresponding effective learning rate plots.

We consistently observe that (1) a higher Bulk-LR positively correlates with increased training speed, and (2) momentum and adaptive learning rates in Adam amplify Bulk-LR, resulting in a larger Bulk-LR compared to Dom-LR.

Table 2: Mean effective learning rates over the first 1000 steps (numbers in parentheses show standard deviation). Training MLP on MNIST-5k using GD and Adam with (+m) and without (-m) momentum. GD uses a learning rate of 0.01, Adam uses a learning rate of 0.001. Momentum is set to $\beta = 0.9$.

| Method | Mean Dom-LR | Mean Bulk-LR |
|---|---|---|
| GD(-m) | 0.0100 (0.0000) | **0.0100** (0.0000) |
| GD(+m) | 0.0012 (0.0092) | **0.1005** (0.0068) |
| Adam(-m) | 0.3317 (0.0571) | **0.4021** (0.0868) |
| Adam(+m) | 0.0356 (0.0194) | **0.7301** (0.4717) |

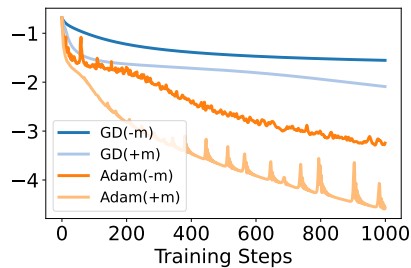

Figure 38: Training loss in log-scale for the experiments in Table 2.

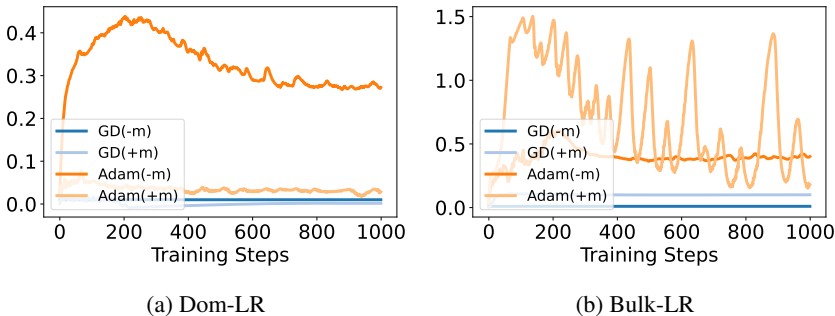

(a) Dom-LR  (b) Bulk-LR

Figure 39: Effective learning rates (smoothed with EMA) for experiments in Figure 2 when training MLP on MNIST-5k.

Table 3: Mean effective learning rates over the first 1000 steps (numbers in parentheses show standard deviation). Training CNN on CIFAR10-5k using GD and Adam with (+m) and without (-m) momentum. GD uses a learning rate of $0.001$, and Adam uses a learning rate of $10^{-4}$. Momentum is set to $\beta = 0.9$.

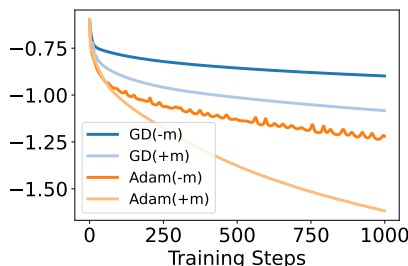

Figure 40: Training loss in log-scale for the experiments in Table 3.

| Method | Mean Dom-LR | Mean Bulk-LR |
|---|---|---|
| GD(-m) | 0.0010 (0.0000) | **0.0010** (0.0000) |
| GD(+m) | 0.0010 (0.0023) | **0.0101** (0.0007) |
| Adam(-m) | 0.0191 (0.0020) | **0.0289** (0.0053) |
| Adam(+m) | 0.0046 (0.0026) | **0.0802** (0.0194) |

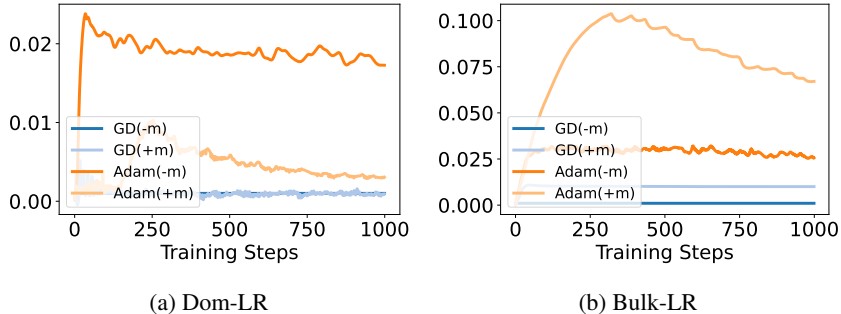

(a) Dom-LR

(b) Bulk-LR

Figure 41: Effective learning rates (smoothed with EMA) for experiments in Figure 3 when training CNN on CIFAR10-5k.

## G  SHARPNESS PLOTS FOR MAIN EXPERIMENTS

In this section, we provide sharpness plots for main experiments when training MLP on MNIST-5k. Figure 42 shows the sharpness plots of Dom-SGD and Bulk-SGD for the experiment in Figure 1a. Figure 43 shows the sharpness plots of Dom-GD and Bulk-GD for the experiment in Figure 9. Figure 44 shows the sharpness plots of Dom-SAM and Bulk-SAM for the experiment in Figure 10.

Quite surprisingly, we observe that Dom-SGD decreases the sharpness, unlike SGD and Bulk-SGD. Moreover, Bulk-SGD increases sharpness more rapidly than SGD. We believe these phenomena are closely interrelated. As Dom-SGD update decreases the sharpness ($\Delta\lambda_{\text{dom}} < 0$), the absence of the dominant component in Bulk-SGD update would lead to larger increase of the sharpness compared to the SGD update ($\Delta\lambda_{\text{SGD}} \approx \Delta\lambda_{\text{dom}} + \Delta\lambda_{\text{bulk}} < \Delta\lambda_{\text{bulk}}$).

Based on these observations, we hypothesize that the dominant component in SGD is primarily an effect of stochastic noise, while the bulk component behaves more similarly to continuous-time gradient flow. The sharpness reduction during Dom-SGD may be closely related to the phenomenon where SGD noise biases optimization toward flatter regions of the loss landscape, as analyzed in prior works (Blanc et al., 2020; Damian et al., 2021; Li et al., 2022). However, these prior analyses focus on dynamics near the minimizer manifold (zero loss), whereas the sharpness reduction in Dom-SGD occurs earlier, before reaching the minimizer manifold.

We hypothesize that the noise in Dom-SGD implicitly reduces sharpness near the manifold of points where the dominant component of the gradient vanishes (i.e., $\chi_k(\nabla L(\theta)) = 0$). Further investigation of this intriguing phenomenon is left as future work.

In the EoS regime, we observe that Bulk-GD increases the sharpness larger than the stability threshold $2/\eta$. Similarly, Bulk-SAM increases the sharpness larger than SAM-edge, the stability threshold of SAM.

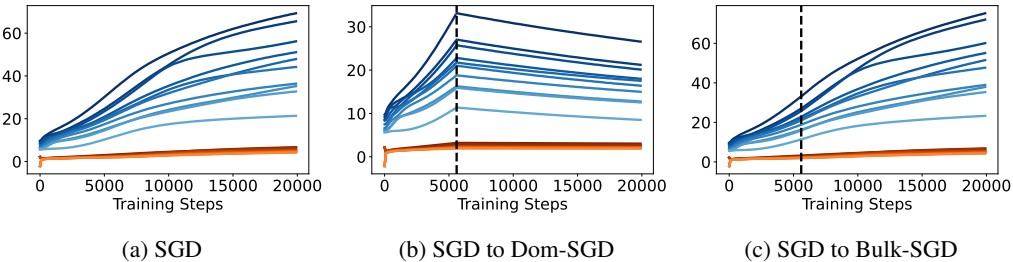

|         (a) SGD          |    (b) SGD to Dom-SGD    |    (c) SGD to Bulk-SGD    |

Figure 42: We train MLP on MNIST-5k using SGD in the GF regime, using a learning rate $\eta = 0.01$ under the same setting as Figure 1a. The plot shows the top eigenvalues of the loss Hessian during SGD and Dom/Bulk-SGD training. The blue curves represent the top-10 eigenvalues, and orange curves represent the next top-10 eigenvalues.

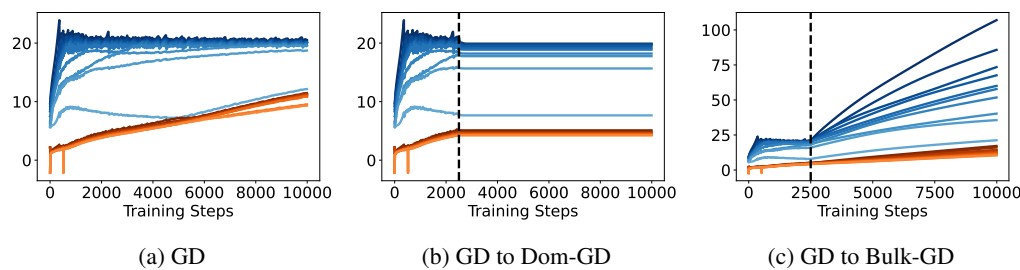

|         (a) GD          |    (b) GD to Dom-GD    |    (c) GD to Bulk-GD    |

Figure 43: We train MLP on MNIST-5k using GD with a large learning rate $\eta = 0.1$ under the same setting as Figure 9. The plot shows the top eigenvalues of the loss Hessian during GD and Dom/Bulk-GD training. The blue curves represent the top-10 eigenvalues, and orange curves represent the next top-10 eigenvalues.

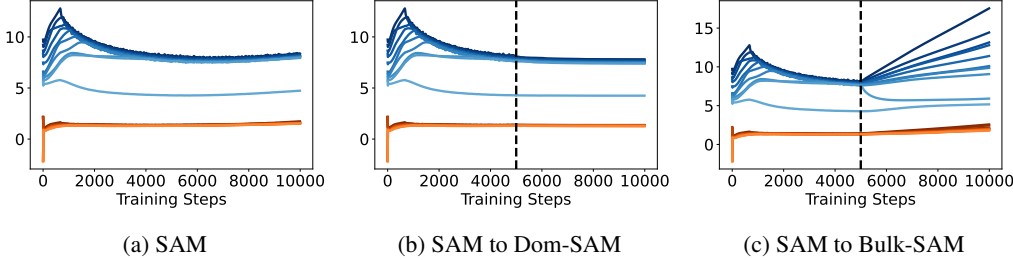

(a) SAM         (b) SAM to Dom-SAM         (c) SAM to Bulk-SAM

Figure 44: We train MLP on MNIST-5k using SAM under the same setting as Figure 10. The plot shows the top eigenvalues of the loss Hessian during SAM and Dom/Bulk-SAM training. The blue curves represent the top-10 eigenvalues, and orange curves represent the next top-10 eigenvalues.

# H   TEST ACCURACY RESULTS FOR DOM-SGD AND BULK-SGD

In this section, we provide preliminary results on test accuracy of SGD, Dom-SGD, and Bulk-SGD. The experiments are conducted on full MNIST dataset with a learning rate $\eta = 0.01$ and the plots of train loss and test accuracy are provided in Figure 45. The results show that the trend of test accuracy is similar to that of train loss, i.e., Bulk-SGD generalizes as well as SGD, while Dom-SGD fails to generalize well. We believe that more systematic experiments beyond this preliminary results would help to provide deeper insights on the generalization characteristics of Dom/Bulk-SGD, and we leave this question as a future work.

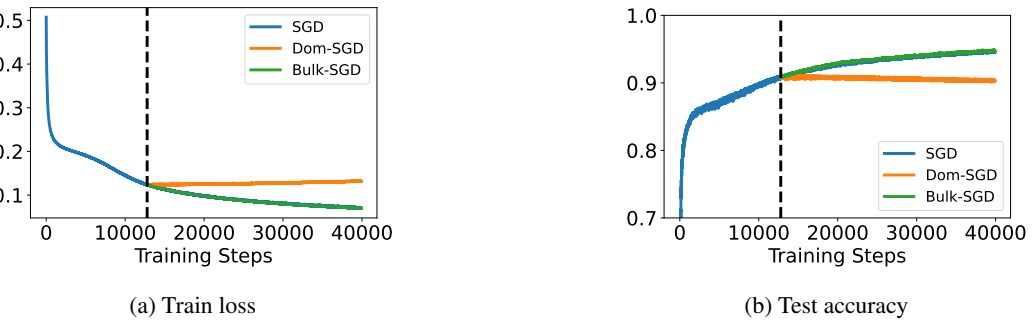

(a) Train loss                  (b) Test accuracy

Figure 45: We train MLP on MNIST using SGD and Dom/Bulk-SGD with a learning rate $\eta = 0.01$. In case of Dom-SGD, training loss does not decrease and test accuracy does not increase. In contrast, for Bulk-SGD, training loss decreases and test accuracy increases similar to SGD.

