# OpenReview forum: "Does SGD really happen in tiny subspaces?"
_ICLR.cc/2025/Conference — ICLR 2025 Poster_

### Official Review · Reviewer_pDDh · 2024-10-17

**Soundness:** 3
**Presentation:** 4
**Contribution:** 2
**Rating:** 6
**Confidence:** 4

**Summary:**

This paper is built on recent studies that claim the gradients during neural network training live in a small subspace spanned by the top few eigenvectors of the Hessian. The authors decompose the gradients into the dominant component that lives in the dominant space and the bulk component that is orthogonal to the subspace. They find that dominant component of the gradient does not contribute to decreasing the training loss, while the bulk component, though only accounts for a small portion of the total gradient magnitude, is the main source driving the reduction in the loss. Through different experiments, the authors demonstrate that this observation holds for SGD, GD on the edge of stability, SAM, and adaptive optimizers. The authors also discuss potential explanations from a loss landscape perspective.

**Strengths:**

The paper is well written with a clear structure and compelling experimental results. The design of the experiments is sophisticated, covering both Dom-SGD and Bulk-SGD setups, along with more nuanced experiments like switching between SGD and GD during training. These additional experiments strongly support the paper’s claims, adding robustness to the argument. Additionally, the observations found in this paper is interesting and should contribute to a better understanding of the training dynamics of deep neural networks.

**Weaknesses:**

Major point:

1.	Generalization: The authors acknowledge in the limitation that this paper does not consider the generalization part. However, in modern deep learning, generalization error is usually more important than training loss. And it is widely believed that SGD has an implicit bias that benefits generalization (arxiv: 1611.03530, 2006.08680, 2306.04251, etc.). The absence of results or discussion on generalization weakens the paper. Including an analysis of generalization effects could significantly enhance the paper's impact.

2.	Bulk-SGD and noise: The results suggest that Bulk-SGD is less noisy than SGD (e. g. in Fig. 1) and Bulk-GD is less noisy than GD (Fig. 9). This reduction in noise may explain the speedup observed in Fig. 9 and 10. On the other hand, it is unclear how this will affect generalization. The authors have not adequately addressed this part in the paper, and I suggest including a more thorough discussion around that.

3.	Dom-SGD and loss increase: It appears that with cross-entropy loss or using a standard architecture (VGG-11) (Fig. 14 and 17), the loss increases if training with Dom-SGD. This is a surprising finding and I think the authors should discuss this phenomenon more explicitly in the main text. Additionally, the absence of training curves for Bulk-SGD in these figures raises questions. Is there a reason for not including the curves for Bulk-SGD? I am also wondering whether this increase in the training loss relates to findings in prior works: arxiv: 2107.11774, 2306.04251.


Minor point:

1.	For results shown in table 1, the authors calculate the effective learning rates over the first 1000 steps. However, the authors also show that the alignment usually happens after some warm-up steps (e. g. Fig. 3). I think it makes more sense to plot the effective learning rates as a function over time (if noise is a concern, then perhaps run some kind of EMA).

2.	Although the toy quadratic model explains the observations well, the source of randomness seems a little contrived. I suggest incorporating the discussion of another model at least in the appendix, where the noise is Gaussian noise directly added on to the gradients, for a more natural comparison.

**Questions:**

Instead of computing the subspace based on the Hessian of the full loss, what if the subspace were computed based on the Hessian of the mini-batch loss. Would alignment still hold? Do the findings in the paper still hold?

---

> ### Author Response · Authors · 2024-11-20
> **Response to Reviewer pDDh**
>
> Thank you for your efforts and your insightful comments! We appreciate that you found **our observations in this paper interesting and should contribute to a better understanding of the training dynamics of deep neural networks**.
>
> Let us address your concerns one by one.
>
> > Generalization
>
> We definitely agree that considering generalization is an important problem. While this question is out of the scope of our paper, we provide preliminary results on the test accuracy of SGD, Dom-SGD, and Bulk-SGD in Appendix H. The additional experiments are conducted on the full MNIST dataset and the plots of train loss and test accuracy for each method are provided. The results show that the trend of test accuracy is similar to that of train loss, i.e., Bulk-SGD generalizes as well as SGD, while Dom-SGD fails to generalize well. We believe that more systematic experiments beyond these preliminary results would be necessary to provide better insights on model generalization, and we leave this question as a future work.
>
>
> > Bulk-SGD and noise / Dom-SGD and loss increase
>
> Thank you for your insightful comments! We appreciate your suggestion and have added a discussion on this phenomenon in the Conclusion section, along with **additional experimental results in Appendix G**. Our experiments consistently show that Bulk-SGD tends to be less noisy than SGD, while Dom-SGD tends to increase the loss over time. Specifically, Dom-SGD exhibits a slow but steady increase in loss in the long run across all experiments, including Figure 1. Additionally, we observe that Dom-SGD decreases sharpness, and we have included sharpness plots for both Dom-SGD and Bulk-SGD.
>
> Interestingly, Bulk-SGD increases sharpness more rapidly than SGD. We believe these phenomena are closely interrelated. As Dom-SGD update decreases the sharpness ($\Delta \lambda_{\text{dom}}<0$), the absence of the dominant component in Bulk-SGD update would lead to larger increase of the sharpness compared to the SGD update ($\Delta \lambda_{\text{SGD}} \approx \Delta \lambda_{\text{dom}} + \Delta \lambda_{\text{bulk}} < \Delta \lambda_{\text{bulk}}$). We currently do not have a fully satisfactory explanation of these phenomena, and we leave this question as a future work.
>
>
> In the additional plots in the Appendix, we focused on verifying our main claim that training cannot occur in the dominant subspace, which is why we did not include the training loss curves for Bulk-SGD. However, we observed the same behavior for Bulk-SGD in those experiments.
>
> Furthermore, we are uncertain whether the prior works demonstrating that SGD can converge to local maxima (arXiv:2107.11774) and the stochastic collapse of SGD (arXiv:2306.04251) are directly related to the loss increase observed in Dom-SGD. Specifically, we do not believe that the loss increase in Dom-SGD is caused by convergence to a local maximum. Could you elaborate further on the potential connection between our findings and the stochastic collapse?
>
>
> > Plot the effective learning rates as a function over time
>
> Thank you for the suggestion. In Appendix F.2, we added the plots of effective learning rates as a function over time for the experiments training MLP on MNIST-5k, CNN on CIFAR10-5k, and Transformer on SST2-1k. Notably, we observe that Adam (with momentum) tends to gradually increase Bulk-LR during training Transformer on SST2-1k (see Figure 37). This observation further supports the idea that Adam effectively adapts to the loss landscape, enabling efficient optimization of Transformers.
>
> > Toy quadratic model with Gaussian noise
>
> Thank you for the suggestion. We added the experiments on the toy quadratic model with Noisy GD (i.e., Gaussian noise) in Appendix D.6. We implement Noisy GD by injecting a noise sampled from an isotropic Gaussian distribution after each GD update. The results are similar to the original toy model. Both GD and Noisy GD trajectories closely follow the y-axis. However, the gradient aligns with the dominant subspace (x-axis) for Noisy GD, unlike GD. Note that we also provide experiment results with Noisy GD under deep learning setups in Appendix D.4, Figure 24.
>
> > Subspace computed based on the Hessian of the mini-batch loss
>
> We compute the subspaces with the full loss Hessian because Gur-Ari et al. (2018) observed that full gradients align with the top eigenspace of the full loss Hessian during SGD training. Moreover, they also observed that the top eigenspace of full loss Hessian (i.e., dominant subspace) is mostly preserved over the long period of training. In contrast, the top eigenspace of mini-batch loss Hessian is unstable and quickly changes over time during training. Therefore, the alignment between gradient and top eigenspace of mini-batch loss Hessian is also unstable compared to that of full loss Hessian.

---

> > ### Comment · Reviewer_pDDh · 2024-11-20
> >
> > I thank the authors for their thoughtful response and for conducting additional experiments to address my comments. I intend to increase my score by 1 to reflect the fact that the authors have satisfactorily addressed most of my concerns. Below, I provide additional comments and suggestions based on the new results and discussions.
> >
> > The reason I inquired about generalization results is that the observations in the paper suggest __the dominant component in SGD likely originates from noise__. This aligns with the perspective offered by the toy model as well. Consequently, I suspect that the noise associated with the dominant component could have a similar effect to the implicit bias of SGD.
> >
> > I also would like to make a comment on the decrease of sharpness observed in appendix G. This finding is consistent with the well-established notion that SGD noise biases optimization towards flatter regions of the loss landscape. This phenomenon has been theoretically supported by prior works (e.g., arXiv:2106.06530), which show that noise in SGD acts as an implicit regularizer on sharpness. I believe this perspective could help contextualize the findings in Appendix G, providing a clearer explanation of the observed trends.
> >
> > Based on these points and results, I *__hypothesize__* that __the dominant component in SGD is primarily an effect of noise__, while the bulk component behaves more like a full-batch gradient descent. If this hypothesis holds, the increase in loss observed with dom-SGD could be explained by stochastic collapse, a phenomenon directly linked to the noise effect. One direct way to test this is to run the experiments (e. g. in Fig. 17) much longer and check if the loss would saturate at some fixed value instead of increasing indefinitely. I strongly suspect that the observation of loss increasement reflects slow convergence toward a local maximum, driven by the noise in dom-SGD.
> >
> > Although these points are largely speculative, I encourage the authors to consider incorporating these insights into the paper if they find them relevant and agree with the interpretations. I am happy to discuss these ideas further if needed.

---

> > > ### Author Response · Authors · 2024-11-21
> > >
> > > We deeply appreciate your thoughtful feedback and insightful comments. We noticed that your score remains unchanged, and we understand this might be due to the lack of an option to assign a score of 7. It would mean a great deal to us if, during the discussion period, you could communicate to AC that you feel the score lies somewhere between 6 and 8.
> > >
> > > Additionally, we greatly value your interpretation that the dominant component in SGD is primarily an effect of noise, while the bulk component behaves more like full-batch gradient descent. This aligns well with our findings, and we are grateful for sharing your perspective. Specifically, the decrease in sharpness during Dom-SGD appears to be closely related to the sharpness reduction effect of SGD. However, we note that prior works (Blanc et al., 2020; Damian et al., 2021; Li et al., 2022) primarily analyzed SGD dynamics near the minimizer manifold (zero loss), while the sharpness reduction effect of Dom-SGD occurs earlier, before reaching zero loss.
> > >
> > > Given this distinct scenario, we hypothesize that the noise in Dom-SGD implicitly reduces sharpness near the manifold of points where the dominant component of the gradient vanishes (i.e., $\chi_k(\nabla L(\theta)) = 0$). We have revised the paper incorporating a discussion on the sharpness reduction effect of Dom-SGD, reflecting your insights, in Appendix G.
> > >
> > > Furthermore, to investigate whether the increase in loss during Dom-SGD might be explained by stochastic collapse, we could design an experiment to directly examine this effect by measuring the number of activated neurons during Dom-SGD. While this is a plausible avenue for future research, we chose not to include this hypothesis in the manuscript to maintain focus and concreteness.
> > >
> > > Regarding the "local maximum" hypothesis, we believe this is unlikely in the context of neural network optimization landscapes. Specifically, a local maximum cannot exist unless the hidden layer representations in the last hidden layer are uniformly zero. To elaborate, if we restrict the parameter space to the parameters of the final linear readout layer, the loss function within this subspace is convex, meaning the block Hessian with respect to the linear layer is PSD. For the network to have a local maximum, this block Hessian would need to be zero, which would occur only if the representation vectors of all data points are zero.
> > >
> > > Blanc et al. (2020), Implicit regularization for deep neural networks driven by an Ornstein-Uhlenbeck like process, COLT 2020.
> > >
> > > Damian et al. (2021), Label Noise SGD Provably Prefers Flat Global Minimizers, NeurIPS 2021.
> > >
> > > Li et al. (2022), What Happens after SGD Reaches Zero Loss? --A Mathematical Framework, ICLR 2022.

---

> ### Comment · Reviewer_pDDh · 2024-11-21
>
> I thank the authors for their further responses. I intend to update the score to 7.

---

### Official Review · Reviewer_LPQC · 2024-10-27

**Soundness:** 3
**Presentation:** 3
**Contribution:** 3
**Rating:** 8
**Confidence:** 4

**Summary:**

This paper challenges existing assumptions about stochastic gradient descent (SGD) learning within a low-dimensional dominant subspace defined by the top eigenvectors of the Hessian of the loss. The authors find empirically that while gradients appear to align with this dominant subspace during training, projecting updates solely into this subspace (Dom-SGD) halts learning. Instead, projecting updates onto the orthogonal "bulk" subspace (Bulk-SGD) is as effective as standard SGD, suggesting that meaningful learning occurs outside the dominant subspace.

**Strengths:**

The paper addresses a question of significant interest in the ML community regarding SGD's effectiveness in low-dimensional subspaces. It helps address potential misconceptions arising from the well-cited work of Gur-Ari et al. (2018), which suggests that gradient descent would occur primarily within a tiny subspace.

A convincing quadratic toy model effectively reinforces the authors’ interpretation of the empirical results, lending credibility to their main conclusions.

The paper is well-structured, with a logical flow that makes the argument easy to follow and the findings readily accessible to readers.

Gur-Ari et al. (2018): https://arxiv.org/abs/1812.04754

**Weaknesses:**

Although the paper includes experiments on three datasets, training is restricted to small subsets (e.g., 5,000 of 50,000 samples of CIFAR10) and primarily uses mean squared error loss instead of cross-entropy, despite focusing on classification tasks. This may limit the generalizability of the findings and should be communicated more clearly.

The paper exclusively examines the effects on training loss, with no analysis of test accuracy under Dom-SGD and Bulk-SGD. Including at least one plot of test performance would offer valuable insights into the practical implications for readers.

**Questions:**

Why did the authors limit their experiments to a small subset of the datasets and can the authors share any insights for training with the full dataset?

Could the authors provide any test accuracy results for Dom-SGD and Bulk-SGD to offer some insight into their impact on model generalization? When Bulk-SGD performs similarly to SGD in terms of training loss, does this trend hold for test accuracy as well?

---

> ### Author Response · Authors · 2024-11-20
> **Response to Reviewer LPQC**
>
> Thank you for your support and for your valuable feedback! We appreciate that you found our work **addresses a question of significant interest in the ML community regarding SGD's effectiveness in low-dimensional subspaces**.
>
> We would like to address your questions as below.
>
> > Why did the authors limit their experiments to a small subset of the datasets?
>
> Let us explain why we decided to conduct experiments on small subsets. Conducting extensive systematic experiments on full datasets is computationally challenging given that Dom-SGD and Bulk-SGD require computing top-10 eigenvalues and eigenvectors of the full loss Hessian at every single iteration. Existing works such as Cohen et al. (2021) did **not** compute the sharpness of full loss Hessian for the CIFAR10 dataset (see https://github.com/locuslab/edge-of-stability). Instead, they computed the proxy of sharpness by randomly selecting a batch of 5k samples and computing the top eigenvalue of mini-batch loss Hessian. However, such an approximation method could lead to an unstable approximation of eigenvectors, making it inappropriate to implement Dom-SGD and Bulk-SGD. For this reason, we chose to perform experiments on smaller subsets of datasets.
>
> To demonstrate that the trends observed on the subsampled datasets are consistent with those on the full dataset, we added a new set of experiments using the full MNIST dataset in Appendix H. The results confirm that our main observations hold true in this setting as well.
>
>
> > Could the authors provide any test accuracy results for Dom-SGD and Bulk-SGD to offer some insight into their impact on model generalization? When Bulk-SGD performs similarly to SGD in terms of training loss, does this trend hold for test accuracy as well?
>
> Thank you for the suggestion. We provide preliminary results on the test accuracy of SGD, Dom-SGD, and Bulk-SGD in Appendix H. As noted above, the additional experiments are conducted on **full** MNIST dataset and the plots of train loss and test accuracy for each method are provided. The results show that the trend of test accuracy is similar to that of train loss, i.e., Bulk-SGD generalizes as well as SGD, while Dom-SGD fails to generalize well. We believe that more systematic experiments beyond these preliminary results would be necessary to provide better insights on model generalization, and we leave this question as a future work.
>
> > Primarily uses MSE Loss instead of cross-entropy, despite focusing on classification tasks.
>
> The choice of MSE loss was intended to clearly distinguish the settings between the small LR (GF) regime and the large LR (EoS) regime. When cross-entropy loss is used with a large LR, progressive sharpening initially occurs and enters the EoS regime, but training does not remain in the EoS regime till the end since sharpness decreases at the end of training (Cohen et al., 2021). In contrast, when MSE loss is used with a large LR, sharpness does not decrease after entering the EoS regime. Therefore, in order to conduct systematic experiments investigating the distinct mechanism of spurious alignment depending on LR, we decided to primarily use MSE loss.

---

> > ### Comment · Reviewer_LPQC · 2024-11-21
> >
> > I thank the authors for their detailed responses and the additional experiments on the full MNIST dataset. While systematic experiments on more complex datasets would further broaden the impact, their current results provide compelling insights into SGD dynamics. I intend to keep my score.

---

### Official Review · Reviewer_rVK6 · 2024-11-01

**Soundness:** 2
**Presentation:** 2
**Contribution:** 1
**Rating:** 3
**Confidence:** 5

**Summary:**

This paper investigates that neural nets cannot be trained within the dominant subspace (i.e., along sharp directions).

**Strengths:**

Many recent papers (Blanc et al., 2022; Li et al., 2022) highlight that the dynamics in bulk subspace (i.e., along flat directions) are crucial for SGD/SAM to move to flat minima, thereby improving generalization.
In contrast, this paper emphasizes the significant role of the dynamics in bulk subspace (i.e., along flat directions) in relation to optimization.

**Weaknesses:**

My primary concern are (i) the paper does not sufficiently explain the main finding, i.e., why only the dynamics in the bulk subspace are crucial for optimization, and (ii) the novelty of many contexts:

- Section 4. This section focus on the alignment between the stochastic gradient and the sharp directions. However,

  - This section fails to adequately explain the main finding: why only the dynamics in the bulk subspace are crucial for optimization, expect for a very toy model.

  - Even regarding the alignment between stochatic gradient and the loss landscape, there are numerous previous works that address this point: $\mathbb{E}[g_tg_t^T]\approx 2 L(\theta)\nabla^2 L(\theta_t)$ (Wojtowytsch, 2021; Mori et al., 2022; Wu et al., 2022; Wang and Wu., 2023), implying that stochastic gradient concentrate more along sharp directions of the landscape (i.e., the bump directions). It appears that the authors have overlooked these works.

- Section 5. This section demonstrates that the gradient of GD aligns with the sharp directions in Edge of Stability (EoS), However,

  - The authors do not explain the main finding: why only the dynamics in the bulk subspace are crucial for optimization.

  - Regarding the alignment between the gradient and sharp directions, this has beed well studied in (Arora et al., 2022; Lyu et al; 2022; Damian et al., 2023) that this occurs due to approximate power iterations). However, the authors do not provide new insights.

**Questions:**

- The experiments focus on the optimization comparsion between SGD, Bulk-SGD, and Dom-SGD.
What are the generalization differences among these methods, particularly in the experiments conducted on MNIST and CIFAR, as shown in Figure 1?

- Could the authors provide sufficient theoretical explanation for the main finding?

---

> ### Author Response · Authors · 2024-11-20
> **Response to Reviewer rVK6 (1/2)**
>
> We appreciate your efforts and valuable feedback. Before addressing each of the reviewers' comments, we would like to highlight the novel contributions of our paper. As Reviewer LPQC noted, our work tackles a potential misconception arising from the influential work of Gur-Ari et al. (2018), which suggests that gradient descent would occur primarily within the dominant subspace. Contrary to this expectation, our empirical findings reveal that projecting each gradient update onto the dominant subspace does not, in fact, decrease the loss—a surprising observation, especially given that gradient vectors appear aligned with the dominant subspace during SGD training. Furthermore, we deepen our analysis of this phenomenon through systematic experiments on real-world datasets, as well as a toy model, providing further insights into the underlying mechanism.
>
> Let us address your concerns one by one.
>
>
> > (Section 4) This section fails to adequately explain the main finding: why only the dynamics in the bulk subspace are crucial for optimization, except for a very toy model.
>
> The primary focus of our paper is to identify and analyze a novel phenomenon in deep learning through systematic experiments. The toy model serves as a tool to provide intuition for the counterintuitive phenomenon we uncovered. Specifically, our ill-conditioned quadratic toy model, though simple, captures the essence of our empirical findings. This is evidenced by its ability to replicate all the key phenomena observed in the deep learning setup (see Section 4.2 for details).
>
> Building on the intuitions from the toy model, we hypothesize that the "ill-conditioned valley" loss landscape of neural networks explains why the bulk subspace dynamics are crucial for optimization (see Section 4.3 for details). In such a landscape, the SGD trajectory remains near the bottom of the valley. The valley is elongated along the bulk subspace, and true optimization progress occurs within this subspace, as the bottom of the valley descends in that direction.
>
>
> > (Section 4) Even regarding the alignment between stochastic gradient and the loss landscape, there are numerous previous works that address this point: $\mathbb{E}[g_tg_t^\top]\approx 2L(\theta)\nabla^2 L(\theta_t)$ (Wojtowytsch, 2021; Mori et al., 2022; Wu et al., 2022; Wang and Wu., 2023), implying that stochastic gradient concentrate more along sharp directions of the landscape.
>
> First, we **did not** compute the alignment between the "stochastic" gradient and the loss landscape. Instead, we measured the alignment between the "full-batch" gradient and the loss landscape along the SGD trajectory. The previous works you mentioned (e.g. Wu et al., 2022) demonstrate that the stochastic noise has a larger bias in sharp directions, but this does not imply that full-batch gradient aligns with top eigenspace because full-batch gradients are "noiseless."
>
> In fact, the "spurious alignment" also occurs even when the stochastic noise is isotropic, i.e., the same bias along every direction. In Appendix D.4, we run Noisy GD in neural network experiments by injecting noise sampled from an isotropic Gaussian distribution after each GD update iteration and observe the alignment between full-batch gradient and top eigenspace.
>
> > (Section 5) The authors do not explain the main finding: why only the dynamics in the bulk subspace are crucial for optimization. The alignment between the gradient and sharp directions has been well studied in (Arora et al., 2022; Lyu et al; 2022; Damian et al., 2023) that this occurs due to approximate power iterations. However, the authors do not provide new insights.
>
> We agree that the alignment between the gradient and dominant subspace in the EoS regime can be understood through approximate power iteration (also known as self-stabilization). Indeed, we cited Damian et al. (2023) and explicitly highlighted its relevance in Remark 2 of Section 5, as well as in the Related Work section. However, we emphasize that the primary contribution of Section 5 is not the discovery of this alignment but rather our observations on Dom-GD and Bulk-GD in the EoS regime. It is important to note that while Damian et al.'s analysis explains the behavior of GD, it does not extend to Bulk-GD. Specifically, their theory cannot predict or explain the training dynamics when the self-stabilization effect, arising from oscillations along the dominant subspace, is absent.
>
> Furthermore, we hypothesize that the "ill-conditioned valley" loss landscape of neural networks explains why dynamics in the bulk subspace are critical for optimization, even in the EoS regime. Specifically, in the EoS regime, the gradient descent (GD) updates jump across the valley, oscillating within the dominant subspace, while the "true" optimization progress occurs within the bulk subspace, where the bottom of the valley descends. This provides an intuitive explanation for why bulk subspace dynamics play a crucial role in optimization.

---

> ### Author Response · Authors · 2024-11-20
> **Response to Reviewer rVK6 (2/2)**
>
> > Q1. The experiments focus on the optimization comparsion between SGD, Bulk-SGD, and Dom-SGD. What are the generalization differences among these methods, particularly in the experiments conducted on MNIST and CIFAR, as shown in Figure 1?
>
>
> Although our paper primarily focuses on optimization, we provide preliminary results addressing the reviewer’s question about generalization differences among SGD, Dom-SGD, and Bulk-SGD in Appendix H. These experiments were conducted on the full MNIST dataset, and we include plots of training loss and test accuracy for each method. The results indicate that test accuracy trends closely mirror those of training loss: Bulk-SGD generalizes as well as SGD, while Dom-SGD struggles to generalize effectively.
>
>
> > Q2. Could the authors provide sufficient theoretical explanation for the main finding?
>
> Our paper is an empirical work that uncovers an interesting and counterintuitive phenomenon in NN training dynamics and proposes (through a toy example) a novel hypothesis that can explain it. We believe that providing a theory on our empirical findings beyond the toy model is definitely an important future research direction, but is currently out of the scope of this paper.
>
> Scientific progress is driven by discovery, the formulation of hypotheses, and their subsequent validation. In our work, we have presented an intriguing discovery and hypothesis, which we believe constitutes a meaningful contribution. While we acknowledge that a full theory is not provided, we feel that rejecting the paper solely on this basis is overly harsh. We kindly ask you to reconsider your evaluation of our manuscript.

---

> ### Author Response · Authors · 2024-11-28
> **Any Remaining Concerns?**
>
> Dear Reviewer rVK6,
>
> Thank you for your initial review of our paper. We wanted to check if you have had the chance to read our rebuttal, as it has been a week since we uploaded our response. The discussion period is coming to an end, and we would greatly appreciate your thoughts on whether our responses resolved the concerns you raised.
>
> If there are any remaining concerns, we would be happy to clarify further. Alternatively, if you find that your concerns have been sufficiently addressed, we kindly ask you to consider adjusting your score accordingly.
>
> Thank you for your time and contributions to the review process.
>
> Sincerely,
>
> Authors

---

> ### Comment · Reviewer_rVK6 · 2024-12-02
>
> I thank the authors for their clarifications; however, my primary concerns remain unresolved.
>
> - **Main concern**: the paper does not sufficiently explain the main finding, i.e., **why only** the dynamics in the bulk subspace are crucial for optimization? While I strongly agree that toy models can be valuable for early-stage research, it is essential that they **preserve the key characteristics of the problem**. For example, neural network training is well-known to occur at the edge of stability (EoS). **However, the toy model used** in this work—a quadratic function—does not exhibit EoS, as it only allows for two states: convergence or divergence.
>
> - Regarding **Section 4**:  I find this section particularly confusing. As clarified, this work measures the alignment of the **full-batch gradient** with the loss landscape. Why, then, does it emphasize the critical role of **stochastic gradients** (e.g., the title of Section 4.1)? I would appreciate **clarification on this apparent contradiction**.
>
> - Regarding Section 5: While I acknowledge the clarification on the distinctions from related works, the current explanation fails to elucidate the underlying mechanism of Bulk-GD.
>
> - Additionally, a recent paper should be discussed.
>
> Given these points, I will maintain my rating.
>
> [1] Cohen et al. Understanding Optimization in Deep Learning with Central Flows. https://arxiv.org/pdf/2410.24206

---

> > ### Author Response · Authors · 2024-12-02
> >
> > We thank the reviewer for their feedback. We would like to emphasize that our primary contribution is a novel empirical observation, which we believe is a meaningful discovery in its own right. Below, we address your concerns one by one.
> >
> >
> > > Main Concern: Insufficient Explanation of the Main Finding
> >
> > We disagree with the reviewer’s concern that our work fails to explain the main finding. As we already outlined in our initial response, the "ill-conditioned valley" loss landscape perspective provides an intuitive explanation for why the bulk component is critical in optimization across both the Gradient Flow (GF) and Edge of Stability (EoS) regimes:
> >
> > 1. **GF regime:** In this regime, (S)GD follows continuous-time gradient flow closely. The dominant components are already "optimized" due to the geometry of the valley, meaning the primary optimization progress occurs in the bulk subspace. The toy quadratic model effectively captures these dynamics by showing that stochastic gradients induce spurious alignment along the dominant subspace even though the bulk subspace drives optimization.
> >
> > 2. **EoS regime:** In this regime, (S)GD oscillates within the dominant subspace due to the self-stabilization effect. However, optimization progress still occurs in the bulk subspace (because the iterates are stable along this subspace), where the bottom of the valley descends.
> >
> > Together, these insights provide a unified view of why bulk component is important in optimization across both regimes. We believe that providing an end-to-end theorem statement is a very challenging task well beyond the scope of our paper. If there is any aspect in our explanation that you find unconvincing, we would greatly appreciate it if you could elaborate on it more.
> >
> > > Toy model is a quadratic function, which does not exhibit EoS
> >
> > Our toy quadratic model is designed to explain the dynamics of neural networks specifically within the **GF regime**, capturing how SGD exhibits spurious alignment in this setting. Although the toy model does not replicate EoS behavior, it effectively captures and replicates the key characteristics (i.e., Phenomena 1--3) of the GF regime observed in neural network training.
> >
> > > Section 4: Role of Stochastic Gradients
> >
> > As we clarified in the previous rebuttal, we **measured the alignment** between **full-batch gradient** and the dominant subspace along SGD trajectories (recall that **SGD trajectories** involve **stochastic gradients**). The role of stochastic gradients of SGD is introducing small perturbations from the GD trajectory. This small perturbations, particularly along dominant directions, induces the spurious alignment when the loss landscape is ill-conditioned.
> >
> > Below, let us elaborate how the stochastic noise of SGD induces the spurious alignment of full-batch gradients through the toy model and ill-conditioned valley loss landscape perspective.
> >
> > 1. **Toy quadratic model:** Consider the ill-conditioned quadratic loss $L(x, y) = \frac{1}{2}(1000x^2 + y^2)$. Due to its ill-conditioned nature, even small stochastic noise in the $x$-direction during SGD causes spurious alignment between the full-batch gradient and the dominant subspace. For example, at $\theta = (0.01, 0.5)$, a slight deviation from the $y$-axis leads to $\chi_1(\nabla L) \approx 0.999$, whereas GD remains at the $y$-axis where $\chi_1(\nabla L) = 0$.
> >
> > 2. **Ill-conditioned valley:** We proposed that neural network training loss exhibits "ill-conditioned valleys" loss landscapes. Given this perspective, GD iterates remain at the valley bottom, where gradients do not align with the dominant subspace. In contrast, stochastic noise in SGD causes slight deviations from the valley bottom, and the high curvature along dominant directions amplifies this, leading to spurious alignment of the full-batch gradient with the dominant subspace.
> >
> >
> > > Section 5: Mechanism of Bulk-GD in the EoS Regime
> >
> > We agree that this is an important direction for future research. We note that our ill-conditioned valley perspective provides a meaningful starting point for understanding why bulk dynamics are critical in this regime. In the EoS regime, sharpness dynamics cause GD updates to oscillate within the dominant subspace, effectively "jumping" across the valley. Meanwhile, optimization progress occurs in the bulk subspace, where the valley’s bottom descends. This separation between oscillations in the dominant subspace and progress in the bulk subspace explains the effectiveness of Bulk-GD in the EoS regime.

---

> > ### Author Response · Authors · 2024-12-02
> >
> > > Related Work
> >
> > Thank you for pointing out the recent paper (https://arxiv.org/pdf/2410.24206). We did not include it in our draft as this work appeared after the ICLR submission deadline. We'll take a careful look.
> >
> > We'd like to note that our ill-conditioned valley perspective was echoed by another recent paper (https://arxiv.org/pdf/2410.05192) that appeared after the ICLR submission deadline. The fact that a similar perspective arose in a different context further supports the credibility of our explanation.
> >
> > This paper studies the Warmup-Stable-Decay (WSD) learning rate scheduler and proposes a "river valley" loss landscape perspective, which shares a similar intuition with our "ill-conditioned valley" model. As the name suggests, a river valley landscape is characterized by steeply sloping hillsides with a river running along the bottom of the valley. Notably, the "river" corresponds to the "bulk" component, while the "hill" corresponds to the "dominant" component in our paper. Both papers emphasize that the fundamental progress in optimization is driven by the river (i.e., bulk) component. However, the motivations differ significantly, as this paper uses the "river valley" perspective to explain the non-traditional loss curve behavior of WSD, while our work uses the "ill-conditioned valley" loss landscape to explain the spurious alignment of SGD and our interesting observations on Dom- and Bulk-GD/SGD dynamics.
> >
> > We will ensure that a discussion of these works is included in the related work section of the revised manuscript.

---

### Official Review · Reviewer_cwmc · 2024-11-02

**Soundness:** 3
**Presentation:** 3
**Contribution:** 3
**Rating:** 6
**Confidence:** 4

**Summary:**

This paper investigates whether neural networks can be effectively trained by focusing only on the "dominant subspace", which is the top eigenspace of the Hessian of the loss, where the gradient seems to align with. The authors projects SGD updates onto this dominant subspace and find that cannot reduce the training loss further. In contrast, removing the dominant subspace from the updates proves equally effective for training, even though it projects out most of the original gradient component. The authors suggest that the alignment between the gradient and the dominant hessian subspace is "spurious". This "spurious" alignment appears consistently across various practical settings, such as the Edge of Stability, Sharpness-Aware Minimization, and when using momentum or adaptive optimizers. The authors also discuss the possible main causes of the phenomenon, and propose some toy models to understand it.

**Strengths:**

This paper systematically investigates the phenomenon of gradient-Hessian alignment in various optimization algorithms, including SGD, GD in the Edge of Stability (EoS) regime, and the Sharpness-Aware Minimization (SAM) algorithm, with a particular emphasis on the analysis of SGD. Previous studies have primarily focused on understanding full-batch algorithms like GD or Adaptive GD without stochasticity.

Moreover, this work takes an initial step toward understanding the 'spurious' alignment where the actual useful component of the gradient is the non-dominant part. They also claim that batch noise is the cause of this alignment by introducing a toy model to provide theoretical intuition. Though it is not very precise to say the mechanism is totally different between GD and SGD (see weakness), I believe the **"ill-conditioned-valley" view of loss landscape** captures why dom-GD doesn't decrease the loss but bulk-GD does.

**Weaknesses:**

First, the so-called 'spurious alignment' is long observed in EoS literature in my opinion. For example, Damian et al. [2] showed that in the EoS regime, (1) the gradient alignment happens (2) the loss decrement in the EoS regime depends on the constrained trajectory **by projecting out the top eigenspace**. It is exactly the finding listed in section 5.1 of this paper that bulk-GD is as effective as GD. I believe the authors should discuss those related works.

The author may argue that "The mechanisms behind gradient alignment differ between GD with large learning rates and SGD with small learning rates." so their findings are novel. But that is also my primary concern of the paper's conclusion "Stochastic noise is the cause of spurious alignment". I agree that "stochastic noise is part of the cause of spurious alignment", but I don't think the alignment mechanism is intrinsically different between GD and SGD.

In section 4. The authors define the "small LR regime" and "large LR regime" of SGD and claim they are different by setting the threshold of $2/\lambda_1$ of the learning rate. This threshold is for the GD descent lemma to hold. Then the authors claim in the small LR regime, the cause of the spurious alignment is the gradient noise. They 'verify' their findings by switching SGD to GD, and observe that the alignment disappears. However, [1] found that SGD begins to oscillate within the top eigenspace smaller than $2/\lambda_1$. An intuitive explanation for this is **injected gradient noise increases the second order term of the Taylor expansion**, making the loss begin to oscillate **when the learning rate is below $2/\lambda_1$** as the descent lemma predicted. In this case, GD is still stable and descent lemma holds, quickly converging back to the "bottom of the valley". That is why the alignment disappears after switching to GD, since after it's in the "bottom of the valley" the gradient component in the top eigenspace will be very small.

Therefore, my argument is that stochastic noise indeed reduces the threshold of learning rate for oscillation and makes it easier to have spurious alignment. But the mechanism is the same: the self-stabilization effect induced by the gradient(+noise) within the top Hessian subspace. **To test this argument, the author should also calculate the full gradient when "spurious alignment" happens and see if the full gradient aligns with the top eigenspace. Also, the experiments should include various batch sizes to test different levels of gradient noise.** If my conjecture holds, I think this work can be seen as a generalized empirical validation of self-stabilization [2] for SGD, which somehow limits the novelty of this paper. I might also be wrong, so if the conjecture is not correct, I will raise my score.

The author also did some small learning rate GD in Appendix D.1 to corroborate their argument. I wonder what is the learning rate of the "small learning rate"? In the original EoS paper, most of the figures will exhibit EoS due to progressive sharpening when the model is trained for a long time. Also, the authors use the cross-entropy loss, where EoS may not happen due to some margin-maximization effect. What if the authors switch to square loss? I believe when you enter the EoS regime (when trained with enough time),  the gradient alignment phenomenon will also appear.

The authors' toy examples constructed the two loss functions to introduce the gradient noise. However, I don't think this reflects the real-world gradient noise, since the 100xy term is artificially added to introduce gradient components in $x$-direction unless $y = 0$. But I think injecting some Gaussian noise biasing toward $x$-direction may also work for the toy case. It would be great to have some empirical evidence as to why the authors believe the gradient noise has a larger component in the top eigenspace.

[1] Lee, Sungyoon, and Cheongjae Jang. "A new characterization of the edge of stability based on a sharpness measure aware of batch gradient distribution." The Eleventh International Conference on Learning Representations. 2023.

[2] Damian, Alex, Eshaan Nichani, and Jason D. Lee. "Self-Stabilization: The Implicit Bias of Gradient Descent at the Edge of Stability."

**Questions:**

1. Can the authors calculate the full gradient when "spurious alignment" happens? Does that also align with the Hessian's top eigenspace? Or it has a very small similarity?
2. Can the authors experiment with some kind of Gaussian noise for the toy example? And can you empirically show why they believe the stochastic noise has a larger bias in the top-eigenspace?
3. For Appendix D.1, can the authors (1) report the learning rates (2) train with longer time s.t. EoS happens (3) add experiments with MSE loss?

---

> ### Author Response · Authors · 2024-11-20
> **Response to Reviewer cwmc (1/3)**
>
> Thank you for your efforts and valuable feedback. Let us address your concerns and questions below. We look forward to discuss further if anything remains unclear.
>
> First, let us highlight one major update in the revised manuscript.
>
> > Definition of small LR regime and large LR regime
>
> In the revised manuscript, we replace the terms "small learning rate regime" and "large learning rate regime" with "GF regime" and "EoS regime", which are defined in Definition 3. In the GF regime, (S)GD closely follows the continuous-time gradient flow. This happens when the sharpness is below the maximum stable sharpness (MSS), where MSS is $2/\eta$ for GD and lower than $2/\eta$ for SGD. We can observe the increase of the sharpness (i.e., progressive sharpening) and a stable decrease of the loss in the GF regime. In the EoS regime, (S)GD oscillates away from the gradient flow, where we can observe oscillation of the sharpness and unstable decrease (spikes) of the loss.
>
> Notably, in Figure 6, GD training initially occurs in the GF regime, but after the sharpness reaches $2/\eta$, training shifts to the EoS regime. This shows that a single run of GD with a constant LR can occur in both of the regimes. Note that, the gradient does not align with the dominant subspace in the GF regime, but after reaching the EoS regime, the spurious alignment occurs. Hence, we thought the original terms small/large LR regime could be misleading given that both GF and EoS regimes are observed with the same LR.
>
>
> > First, the so-called 'spurious alignment' is long observed in EoS literature in my opinion. For example, Damian et al. [2] showed that in the EoS regime. (1) the gradient alignment happens
>
> You are right, the alignment in the "EoS regime" along the GD trajectory arises from the self-stabilization mechanism by Damian et al. (2023). We did cite Damian et al. and explicitly mentioned its relevance in the paper, in Remark 2 of Section 5 (and also in the Related Work section). We emphasize that the primary contribution of Section 5 is not the discovery of this alignment but rather our observations on Dom-GD and Bulk-GD in the EoS regime.
>
> > (2) the loss decrement in the EoS regime depends on the constrained trajectory by projecting out the top eigenspace. It is exactly the finding listed in section 5.1 of this paper that bulk-GD is as effective as GD.
>
> This is not true, our finding that Bulk-GD is as effective as GD in the EoS regime is distinct from the finding in Damian et al (2023). Note that while Damian et al.'s analysis explains the behavior of GD, it does not extend to Bulk-GD. Their theory cannot predict or explain the training dynamics when the self-stabilization effect, arising from oscillations along the dominant subspace, is absent.
>
> Specifically, the self-stabilization mechanism by Damian et al. considers constrained trajectory on the "stable set", the set of points where gradient and top eigenvector is orthogonal and sharpness is below $2/\eta$. They prove that GD trajectory in the EoS regime closely tracks the projected gradient descent (PGD) trajectory on the "stable set" while oscillating along the top eigenvector. This PGD trajectory is different from Bulk-GD we consider. Bulk-GD projects each gradient update on the bulk subspace, while PGD projects each gradient update on the stable set. As a result, sharpness grows larger than $2/\eta$ for Bulk-GD, while in contrast, sharpness remains at (or below) $2/\eta$ along PGD trajectory by the definition of a stable set. In Appendix G, we provide a sharpness plot of Bulk-GD in Figure 43. Note that Bulk-GD increases the sharpness larger than $2/\eta$.

---

> ### Author Response · Authors · 2024-11-20
> **Response to Reviewer cwmc (2/3)**
>
> > I agree that "stochastic noise is part of the cause of spurious alignment", but I don't think the alignment mechanism is intrinsically different between GD and SGD. (...) Therefore, my argument is that stochastic noise indeed reduces the threshold of learning rate for oscillation and makes it easier to have spurious alignment. However, the mechanism is the same: the self-stabilization effect induced by the gradient(+noise) within the top Hessian subspace.
>
> We definitely agree that stochastic noise reduces the "stable" learning rate (i.e., stable LR of SGD is smaller than stable LR of GD), as found by previous works. However, in the GF regime of SGD, we considered using learning rates smaller than the stable LR of SGD, and hence the mechanism behind the gradient alignment is different from the self-stabilization effect. Let us provide concrete evidence on why this is the case.
>
> 1. Sharpness smoothly increases along the SGD trajectory: In Figure 2, we plot top eigenvalues of Hessian during SGD training. Notice that top eigenvalues smoothly increase (i.e., progressive sharpening occurs). If the SGD training were in the EoS regime, then sharpness should oscillate, as observed in e.g., Lee and Jang (2023). This indicates that experiments we did for SGD in the GF regime use LRs smaller than the stable LR of SGD.
> 2. GD and SGD trajectories are close to each other: In Appendix D.2, we observe that trajectories of SGD (exp in Figures 1,2,3) closely track that of GD (exp in Figures 18,19,20), under the same initialization and LR. If LR were larger than the stable LR of SGD, then SGD trajectory would become unstable and move away from GD trajectory.
>
> > To test this argument, the author should also calculate the full gradient when "spurious alignment" happens and see if the full gradient aligns with the top eigenspace.
>
> We did calculate the full gradient when we computed the gradient alignment in the SGD experiments. In fact, we **did not** measure the alignment between the mini-batch gradient and top eigenspace. Instead, we measured $\chi_k(\nabla L(\theta))$, the alignment between full-batch gradient and top eigenspace along the SGD trajectory (e.g., see Figures 3, 4, 5, 7).
>
> > Also, the experiments should include various batch sizes to test different levels of gradient noise.
>
> Thanks to your suggestion, we conducted additional experiments with various batch sizes and the results are provided in Appendix D.4, Figure 25. We train MLP on MNIST-5k with the same initialization and LR and vary batch sizes by {50, 100, 500, 1000, 5000(full-batch)}. (Here, the alignments are still computed with the full gradients.) We observe that the loss curves are similar to each other. Moreover, a smaller batch size increases the gradient alignment. This observation is consistent with our experiments on Noisy GD in Appendix D.4, Figure 24. These observations further support our finding that noise causes spurious alignment in SGD, and this is different from the self-stabilization effect in the EoS regime.
>
> > Q1. Can the authors calculate the full gradient when "spurious alignment" happens? Does that also align with the Hessian's top eigenspace? Or it has a very small similarity?
>
> As we mentioned above, we did calculate the full-batch gradients and observed that they align with the Hessian's top eigenspace. For example, in Figure 3, we measure $\chi_k(\nabla L(\theta))$, the alignment between the full-batch gradient and dominant subspace, and observe that the value stays near 1, showing that the full-batch gradient aligns with the dominant subspace along the SGD trajectory.

---

> ### Author Response · Authors · 2024-11-20
> **Response to Reviewer cwmc (3/3)**
>
> > Q2. Can the authors experiment with some kind of Gaussian noise for the toy example? And can you empirically show why they believe the stochastic noise has a larger bias in the top-eigenspace?
>
> First, we would like to emphasize that our observation does not necessarily imply that stochastic noise has a larger bias in the top eigenspace, given that we measure the alignment of full-batch gradients, but not mini-batch gradients along SGD trajectories. Indeed, **the "spurious alignment" also arises when we run Noisy GD for neural network training**, by injecting noise sampled from an isotropic Gaussian distribution after each GD update iteration. The results are presented in Appendix D.4. Notice that even when the noise is isotropic (i.e., same bias to every direction), we can still observe the alignment between full-batch gradient and top eigenspace, which happens due to the ill-conditioned landscape.
>
> We also added the experiments on the toy quadratic model with Noisy GD (i.e., Gaussian noise) in Appendix D.6. We implement Noisy GD by injecting a noise sampled from an isotropic Gaussian distribution after each GD update. The results are similar to the original toy model. Both GD and Noisy GD trajectories closely follow the y-axis. However, the gradient aligns with the dominant subspace (x-axis) for Noisy GD, unlike GD. Note that we also provide experiment results with Noisy GD under deep learning setups in Appendix D.4, Figure 24.
>
> > Q3. For Appendix D.1, can the authors (1) report the learning rates (2) train with longer time s.t. EoS happens (3) add experiments with MSE loss?
>
> First, the learning rates of ALL experiments are reported in Appendix B Experimental Details. For experiments in Appendix D.1, the learning rates are reported in Appendix B.3 and here are the values:
> - MLP on MNIST-5k: 0.01
> - CNN on CIFAR10-5k: 0.001
> - Transformer on SST2-1k: 0.001
>
> Note that the learning rates (and also the initialization) for experiments in Appendix D.1 (Figures 18,19,20) and experiments in the main text (Figures 1,2,3) are the same, but the only difference is the choice of optimizer (GD vs SGD).
>
> Second, every experiment in the paper uses MSE loss, except Appendix C.1 (we use cross-entropy loss here).
>
> Finally, if we train longer until EoS happens, then the gradient alignment phenomenon will also appear after entering the EoS regime. This phenomenon can be naturally expected from our experiments at the EoS regime in Section 5. Specifically, if we run GD with a constant LR $\eta$, the sharpness initially increases (progressive sharpening) until it reaches $2/\eta$. This phase of training is in the GF regime, where alignment does not occur. As training continues, GD enters the EoS regime, where sharpness oscillates around $2/\eta$. In the EoS regime, the alignment between gradient and dominant subspace occurs.

---

> > ### Comment · Reviewer_cwmc · 2024-11-20
> >
> > Thanks to the authors for the very detailed response. Many of the responses help clarify some of my misunderstandings about the settings, including the definition of $\chi(\nabla L)$, the learning rate setting, and the MSE loss setting. I apologize for missing the Appendix B setting which should be obvious, and I appreciate the authors' effort in revising the current version of the paper.
> >
> > **Based on the authors' response arguing that the sharpness stably increases instead of oscillating when SGD is in the small learning rate/GF regime**, I believe that the mechanism behind the gradient-hessian alignment in this regime differs from EoS. This is a **novel** observation that does not fall into the previous characterization in my previous review (which is based on the EoS regime understanding). I do prefer the new characterization of the "GF regime" and "EoS regime", which highlights the authors' contribution. Therefore, I increase my score to 5. This is an interesting finding that helps better understand the optimization dynamics in the unstable regime as a supplement to EoS. If the authors could highlight why this regime differs from the EoS regime in the main paper (e.g. include your response with more details and explanations), I would like to increase the score to 6.
> >
> > I cannot give a higher score for now because I would like to see more evidence for the spurious alignment in **GF regime**. The empirical evidence of the paper currently is more biased toward the bulk/outlier GD, which can be a common phenomenon in both GF and EoS regimes (e.g. some early papers explaining the EoS regime for GD already argued this point, like Damian et al, Li et al. [1]). Also, the $\chi(\nabla L)$ change after switching between GD/SGD can happen in the EoS regime for SGD. Therefore, the paper would benefit from stressing more on their novel findings, i.e. the spurious alignment in the GF regime for SGD.
> >
> > A minor opinion on your previous argument on Damian et al., I still believe their projected GD is very similar to bulk-GD. Though this "stable set" is not rigorously equivalent to bulk-GD, the authors can refer to section 4.3 which describes the gradient updates projecting out the top eigenvector $u_t$ and one other direction of the sharpness gradient $\nabla S$. When $\theta$ has a high dimension, it is almost equivalent to bulk-GD. Of course, it does not apply to the authors' characterization of SGD in the GF regime.
> >
> > Some Typos: Line 66 in the caption CIAFAR-10.
> >
> > [1] Damian, Alex, Eshaan Nichani, and Jason D. Lee. "Self-Stabilization: The Implicit Bias of Gradient Descent at the Edge of Stability."
> > [2] Li, Zhouzi, Zixuan Wang, and Jian Li. "Analyzing sharpness along gd trajectory: Progressive sharpening and edge of stability." arXiv preprint arXiv:2207.12678 (2022).

---

> > > ### Author Response · Authors · 2024-11-20
> > >
> > > We greatly appreciate your recognition of the novelty of our work and your insightful suggestions. In response, we have revised the manuscript to better emphasize the distinction between the GF regime and the EoS regime in the main text.
> > >
> > > In Section 4.1, we have expanded the discussion of the GD/SGD switching experiments, noting that "SGD remains in the GF regime as sharpness stably increases, rather than oscillating." Additionally, we clarify that "This highlights that the gradient alignment during SGD is not driven by the self-stabilization effect (Damian et al., 2023) observed in the EoS regime."
> > >
> > > Furthermore, in Remark 5 of Section 5.1, we explicitly highlight the different mechanisms behind gradient alignment in SGD during the GF regime and GD during the EoS regime. We hope these revisions address your concerns and provide a clearer understanding of the distinctions between these regimes.

---

> > > > ### Comment · Reviewer_cwmc · 2024-11-20
> > > >
> > > > Thanks to the authors for the timely response. I increased the score.

---

> > > > > ### Author Response · Authors · 2024-11-21
> > > > >
> > > > > Thank you very much for the thoughtful feedback and suggestions. We truly appreciate it.

---

### Author Response · Authors · 2024-11-20
**Revision of Paper**

Dear reviewers and AC(s),

We express our deepest gratitude for your valuable comments and constructive feedback.

We would like to announce that we made a revision to our submission. For your convenience, we marked newly added/modified sentences, paragraphs, and sections in *orange*.

Some noteworthy changes in this revision include:
- We replace the terms "small learning rate regime" and "large learning rate regime" with "GF regime" and "EoS regime", which are defined in Definition 3. In the GF regime, (S)GD closely follows the continuous-time gradient flow. This happens when the sharpness is below the maximum stable sharpness (MSS), where MSS is $2/\eta$ for GD and lower than $2/\eta$ for SGD. We can observe the increase of the sharpness (i.e., progressive sharpening) and a stable decrease of the loss in the GF regime. In contrast, in the EoS regime, (S)GD oscillates away from the gradient flow, and we can observe oscillation of the sharpness around the MSS and unstable decrease (spikes) of the loss.
- We conducted additional experiments on the full MNIST dataset and provided both train loss and test accuracy results of Dom-SGD and Bulk-SGD. The results are provided in Appendix H. We observe that test accuracy trends closely mirror those of training loss: Bulk-SGD generalizes as well as SGD, while Dom-SGD struggles to generalize effectively.
- We conducted additional experiments and provided sharpness plots of Dom/Bulk-SGD in the GF regime, Dom/Bulk-GD in the EoS regime, and Dom/Bulk-SAM in Appendix G. Interestingly, Dom-SGD decreases the sharpness, while Bulk-SGD increases the sharpness more rapidly than SGD.
- Other additional experimental results include: SGD with varying batch sizes (Appendix D.4), toy model with Gaussian noise (Appendix D.6), and effective learning rates as a function of training steps (Appendix F.2).

We would greatly appreciate it if you could take another look at our revised manuscript and consider updating your scores if needed.

Sincerely,

Authors

---

### Meta-Review · Area_Chair_eUSS · 2024-12-23

**Metareview:**

This paper challenges a previously observed phenomenon that "SGD happens in tiny subspaces." The primary observation is that when the SGD update is projected onto the dominant subspace of the loss Hessian, the training loss does not decrease further. On the other hand, projecting out the dominant subspace is just as effective as the original update, despite removing the majority of the original update component. This paper convincingly demonstrates this phenomenon in a range of experiments across different optimizers and learning regimes (including gradient-flow and edge-of-stability regimes), and provides a toy theoretical example. The paper's contribution is novel, solid, and interesting.

**Additional Comments On Reviewer Discussion:**

3 out of 4 reviewers recommended acceptance after the discussion period. Reviewer rVK6 raised the concern that the paper does not "sufficiently explain" the observed phenomena. The authors' response provided an intuitive explanation based on a toy model and the "ill-conditioned valley" hypothesis, offering insights into both the GF regime and the EoS regime. The AC thinks that this response is satisfactory, and that identifying this phenomenon and empirically verifying it are already significant contributions.

---

### Decision · Program_Chairs · 2025-01-22

Accept (Poster)